# Small design modifications can improve the primary stability of a fully coated tapered wedge hip stem

Katja Glismann[1]*, Tobias Konow[1], Frank Lampe[2], Benjamin Ondruschka[3], Gerd Huber[1], Michael M. Morlock[1]

1 Institute of Biomechanics, TUHH Hamburg University of Technology, Hamburg, Germany, 2 Asklepios Klinik Barmbek, Hamburg, Germany, 3 Institute of Legal Medicine, University Medical Center Hamburg-Eppendorf, Hamburg, Germany

* katja.glismann@tuhh.de

**Data Availability Statement:** All relevant data of the experiments are within the paper and its Supporting Information files. Data concerning specific sizes of the stems can be found in http://

## Abstract

Increasing the stem size during surgery is associated with a higher incidence of intraoperative periprosthetic fractures in cementless total hip arthroplasty with fully coated tapered wedge stems, especially in femurs of Dorr type A. If in contrast a stem is implanted and sufficient primary stability is not achieved, such preventing successful osseointegration due to increased micromotions, it may also fail, especially if the stem is undersized. Stem loosening or periprosthetic fractures due to stem subsidence can be the consequence. The adaptation of an established stem design to femurs of Dorr type A by design modifications, which increase the stem width proximally combined with a smaller stem tip and an overall shorter stem, might reduce the risk of distal locking of a proximally inadequately fixed stem and provide increased stability. The aim of this study was to investigate whether such a modified stem design provides improved primary stability without increasing the periprosthetic fracture risk compared to the established stem design. The established (Corail, DePuy Synthes, Warsaw, IN, US) and modified stem designs (Emphasys, DePuy Synthes, Warsaw, IN, US) were implanted in cadaveric femur pairs (n = 6 pairs) using the respective instruments. Broaching and implantation forces were recorded and the contact areas between the prepared cavity and the stem determined. Implanted stems were subjected to two different cyclic loading conditions according to ISO 7206–4 using a material testing machine (1 Hz, 600 cycles @ 80 to 800 N, 600 cycles @ 80 to 1600 N). Translational and rotational relative motions between stem and femur were recorded using digital image correlation. Broaching and implantation forces for the modified stem were up to 40% higher (p = 0.024), achieving a 23% larger contact area between stem and bone ($R^2$ = 0.694, p = 0.039) resulting in a four times lower subsidence during loading (p = 0.028). The slight design modifications showed the desired effect in this in-vitro study resulting in a higher primary stability suggesting a reduced risk of loosening. The higher forces required during the preparation of the cavity with the new broaches and during implantation of the stem could bare an increased risk for intraoperative periprosthetic fractures, which did not occur in this study.

synthes.vo.llnwd.net/o16/LLNWMB8/US%
20Mobile/Synthes%20North%20America/Product
%20Support%20Materials/Technique%20Guides/
Surgical%20Technique%20Guide%20-%
20Emphasys%20Femoral%20Solutions.pdf and
https://synthes.vo.llnwd.net/o16/LLNWMB8/US%
20Mobile/Synthes%20North%20America/Product
%20Support%20Materials/Technique%20Guides/
DSUSJRC01161350%20Rev%204%20-%
20197186-211129%20DSUS%20CORAIL%
20Primary%20ST%20-%20US%20-%20Brand%
20Update-6.pdf."

**Funding:** Institutional financial support and
provision of implants and instruments was
received by DePuy Synthes. The funders had no
role in study design, data collection and analysis,
decision to publish, or preparation of the
manuscript.

**Competing interests:** MMM is a paid consultant of
DePuy Synthes and obtains research support as a
Principal Investigator from Ceramtec, DePuy, and
Beiersdorf. He obtains speaker's fees from
Aesculap, Ceramtec, DePuy, Zimmer, Peter Brehm,
Corin, and Mathys andis in the editorial board
"Trauma und Berufskrankheit." GH is an associated
member of the board of the German Society of
Biomechanics. F. L. is a paid consultant of Depuy
Synthes and Aesculap. B. O. is a board member of
the German Society of Legal Medicine.

## Introduction

Insufficient initial stability of cementless stems used in total hip arthroplasty (THA) surgery can lead to micromotions at the bone-implant interface exceeding the critical level of about 150 µm [1]. This can result in failure of bone ingrowth into the implant interface and consequent stem loosening [1–3]. Loosening is responsible for up to 22.7% of revision surgeries according to the German Arthroplasty Register (EPRD) [4].

Stem subsidence of a loose stem can result in a periprosthetic fracture (PPF) especially in combination with a low energy trauma [5]. PPFs are present in 24.6% of THA revisions [6] and deemed responsible for the revision in 15.9% of the cases in the EPRD [4]. To target these failure mechanisms the problem of stem loosening needs to be addressed.

The Corail stem (DePuy Synthes, Warsaw, IN, US) is the most frequently implanted uncemented stem in the EPRD with about 38.900 treatments reported in 2022 [4]. This stem has a very good success rate with a 10 year survival of 97.6% for the standard version of the stem for all reasons of revision [7]. However, in two joint registries implant size has shown a significant influence on the risk of loosening with a two to three times higher risk in women and a four to six times higher risk in men when a small stem size is implanted (comparison of Corail sizes 8 and 9 to all larger sized Corail stems implanted) [7, 8].

The increased risk for smaller sizes might be caused by a mismatch between the metaphyseal-diaphyseal geometry of the stem design and the anatomy of the human femur [9, 10]. The surgeon can get the impression of sufficient stem primary stability during implantation due to distal (diaphyseal) contact and fixation of the stem, even so no sufficient stability has been achieved in the proximal (metaphyseal) region [11, 12]. Male patients are more susceptible to this problem, as they are more likely to have femurs of Dorr type A. The narrow diaphysis and wide metaphysis increases the risk of mismatch ("champagne flute shape") [10–12].

Once the distal fixation is reached, surgeons are reluctant to increase the size of the stem, as would be necessary to improve the metaphyseal fixation in the proximal trabecular bone [5], due to fear of an intraoperative periprosthetic fracture. This is facilitated by the non-linear size increments [13], where for example the length of the stem increases by 15 mm between sizes 8 to 9, whereas for sizes larger than size 10, the increase is reduced to 5 mm per size [14]. This situation might prevent the surgeon to use a larger size and result in potential undersizing to avoid fractures during implantation [13].

Precise templating can reduce the amount of incorrect stem sizing and is an absolute necessity [14]. A well-defined workflow during surgery, including mandatory templating and adherence to the preoperative planned size during implantation, was shown to reduce the size-related revisions [9, 15]. However, distal cortical reaming was needed in several cases to allow the implantation of the templated stem size. Distal cortical reaming adds operative time, which endangers the patient [16] and such should not be made the standard solution to this problem.

Alternatively, a modification of the stem geometry with a wider proximal stem can be a solution to reduce the risk of metaphyseal diaphyseal mismatch in specific femurs.

Even for a distally fixed stem, the modified (larger) geometry of the implant in the metaphyseal area with a larger bone-implant interface should be beneficial [17]. The modifications might also alter the local bone loading due to the different wedging effect of the modified broaches and stems [13].

A typical approach to analyze the consequences of these small design modifications would be a finite-element-analysis [18–22]. However, models always require an experimental validation by a biomechanical study. Experiments comparing different stem designs for primary stability [23, 24] and micromotions [25–27] have been conducted in the past while subsidence is mostly analyzed from clinical studies [28–30].

The aim of the given study was to investigate whether small design modifications of an established stem design can provide an improved primary stability without increasing the PPF risk. The design modifications include shortening of the stem and widening of the proximal stem as well as narrowing the stem at the tip. Those modifications are expected to be beneficial especially in femurs of Dorr type A.

## Methods

Six fresh cadaveric femoral pairs of male donors in the age range of 37 to 65 (median 58.5 years, median of lower half ($Q_1$) to median of upper half ($Q_3$) 45 to 64) were excised, anonymized and stored below -20˚C for less than eight months. The Ethics Commission of the Medical Association Hamburg (2023-300374-WF) approved this study. The data were analyzed anonymously and no association between individual patient data and specimens was possible.

Initial CT-scans (120 kV, 0.4 mm slice thickness, Incisive CT 128; Philips, Netherlands) were taken with a calibration phantom (QSA; QRM, Germany). Houndsfield units were converted into bone mineral density (BMD) using Structural Insight 3 as used before (University Hospital Schleswig-Holstein, Germany [31]). The BMD was assessed for a cubic region of interest (10 x 10 x 10 mm) at the center of the femoral head above the Ward's triangle (AVI-ZOLite 9.7.0; Thermo Fisher Scientific, MA, US).

The femurs were thawed at room temperature prior to testing [32]. The distal femur was sectioned 130 mm distally from the templated stem tip position and embedded in metal adapter flanges to a height of 60 mm (Technovit 4004; Kulzer, Germany). This created a consistent distance of 70 mm between embedding and the tip of the implant [33].

Surface models of the broaches as well as the implants were created from laser-scans (scan resolution: 0.2 mm, Handyscan 3D; Creaform, Ametek, PA, US). Preoperative templating (Velys Surgery, DePuy Synthes, Warsaw, IN, US) and implantation were conducted by an experienced senior orthopedic surgeon with more than 1000 implantations using the established stem design.

### Femur preparation and stem implantation

The modified stem design (Emphasys, DePuy Synthes, Warsaw, IN, US; Fig 1B) was compared to the established design (Corail, DePuy Synthes, Warsaw, IN, US; [14, 34]; Fig 1B). Both are fully porous hydroxyapatite-coated (HA) tapered wedge stems. Six stems of each stem type were implanted according to the respective surgical instructions by an experienced surgeon [14, 34], with the exception of using a canal finder rasp during implantation of Corail stems, which is not part of the standard instruments (Fig 1A). Reamers for diaphyseal cortical reaming were provided. The surgeon was instructed to stop at a stem size below the preoperatively planned size when he felt that sufficient fixation of the stem was achieved and that an intraoperative fracture could occur by increasing to the planned size. He decided based on his experience.

Forces were measured in the broach handle and the stem impactor during broaching and implantation (9333 A; Kistler, Germany; Fig 2). Recording of the force data for a duration of 65 ms was triggered by an increase in the force signal above the threshold of 300 N (sample rate of 800 kHz, LabVIEW 2019, National Instruments, Austin, TX, US). A peak length of 0.3 ms for a single stroke was expected.

A laser-scan of the femur with the final broach positioned in the femur was obtained. The broach was then removed and a CT-scan of the femur with the prepared cavity was taken. The stems were implanted using the respective impactor tips. The Emphasys impactor has a

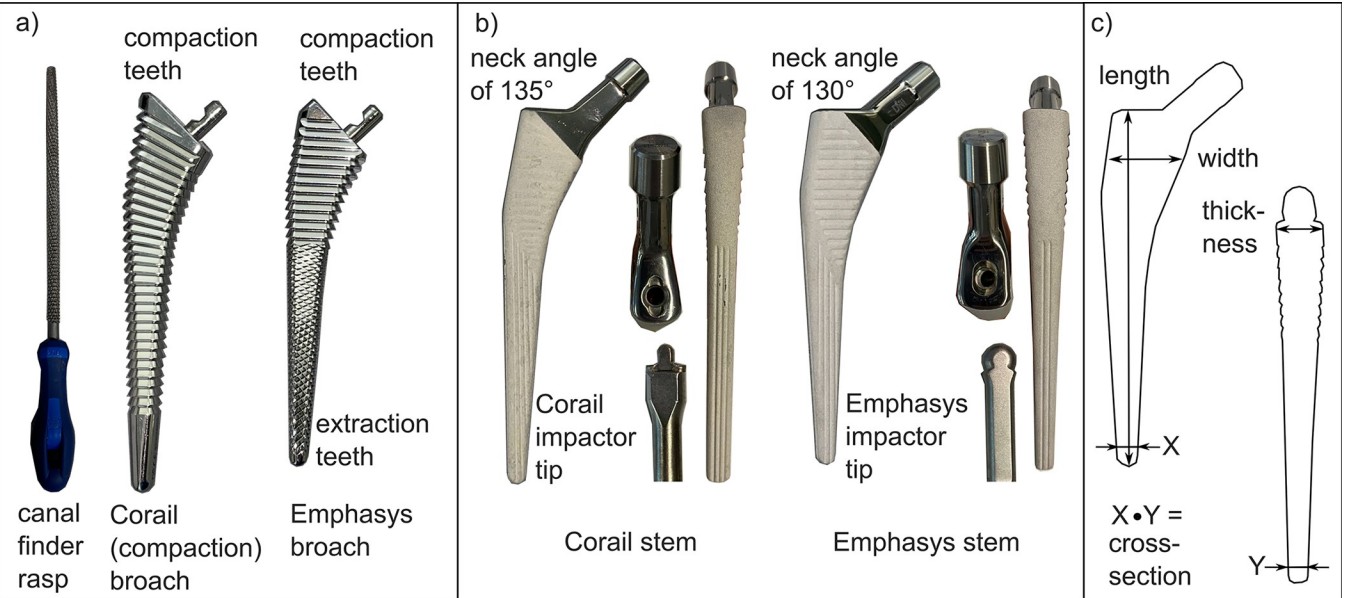

**Fig 1.** (a) The instruments for the preparation of the cavity. From left to right: canal finder rasp similar to [35], Corail compaction broach and Emphasys broach. (b) Stem designs of equal size shown from the side, the top with the corresponding impactor tips, and the back for the Corail and the Emphasys stem. (c) Parameters to quantify design modifications.

rounded tip to increase the angles from which the surgeon can apply force to the stem during implantation (Fig 1B) [34]. Finally, a second laser-scan was taken of the femur with the inserted stem.

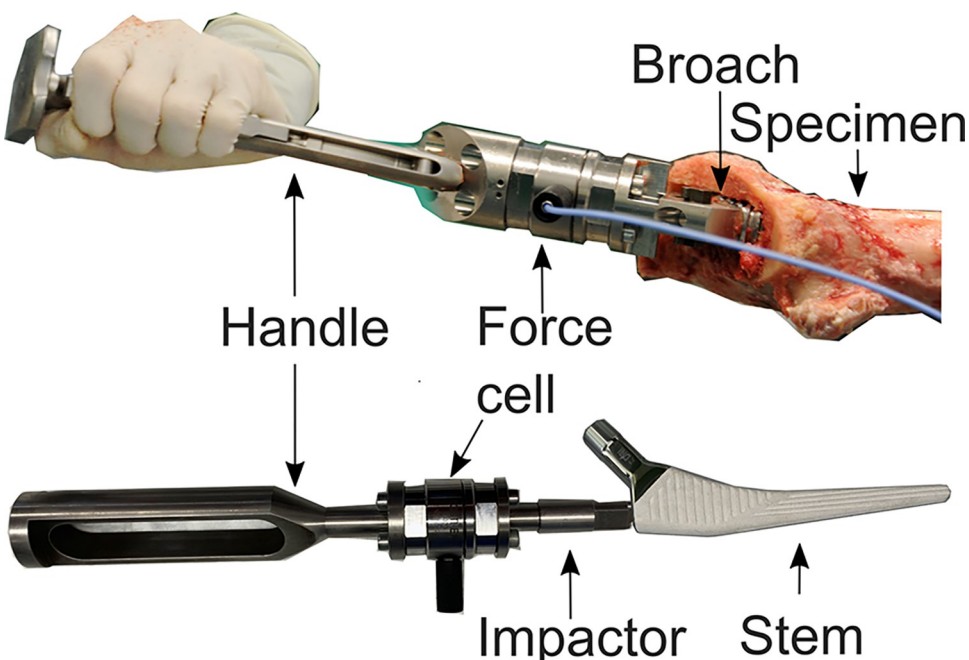

**Fig 2. Position of the dynamic force cell for cavity broaching and stem impaction.**

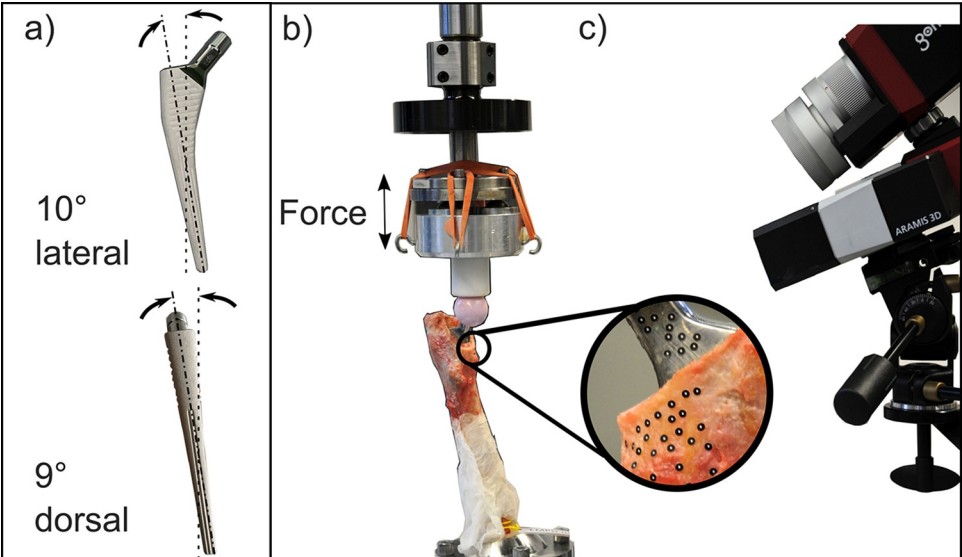

**Fig 3.** (a) Stem alignment according to ISO 7206–4 for (b) cyclic loading in the testing machine using a PE piston against a ceramic head for force application with (c) digital image correlation measurement to determine the relative motion between implant and femur based on markers (Ø 0.4 mm).

The surgeon was asked to comment throughout his impressions of the surgical instruments and the implantation procedure of either stem design.

## Mechanical testing

The implanted stems were aligned according to ISO 7206–4 with a 10˚-degree lateral and 9˚-degree dorsal tilt (Fig 3A) [33]. Sinusoidal dynamic loading with two force levels was applied using a material testing machine under force control (1 Hz; load limits of the force cell 6.3 kN, 300 Nm; Bionix, MTS, Eden Prairie, MN, US; Fig 3B): the lower loading condition consisted of 600 cycles between 80 N to 800 N, the higher of 600 cycles between 80 N to 1600 N. A polyethylene piston served for force application (Fig 3B). Digital image correlation (DIC) with a measurement volume of 100 x 80 x 50 mm was used for contactless analysis of translational and rotational relative motions between bone and implant every 200 cycles for 5 seconds (25 fps, 9 recorded time slots; ARAMIS 3D, MV 100; Carl Zeiss GOM Metrology GmbH, Germany; Fig 3C).

## Data analysis

The nominal press-fit was calculated as the mean distance between the stem surface and the corresponding broach surface (PolyWorks|Inspector 2020; InnovMetric Software, Canada). The respective geometries were derived from laser-scans. These were aligned directionally based on the axes of the distal sections of the stem and broach and positionally by aligning the top surface of the broach (attachment of the handle) with the proximal end of the implant coating (which it corresponds to). The difference in dimensions between corresponding stem sizes of the Corail and the Emphasys design was determined for the length, a corresponding cross-section outside the vertical grooves of either stem and the largest proximal width and thickness of either stem design (Fig 1C).

Dorr types of femurs were determined based on the cortical-canal shape (CCS) calculated from a patient-height-adapted canal-to-calcar ratio (CCR) divided by the cortical index (CI)

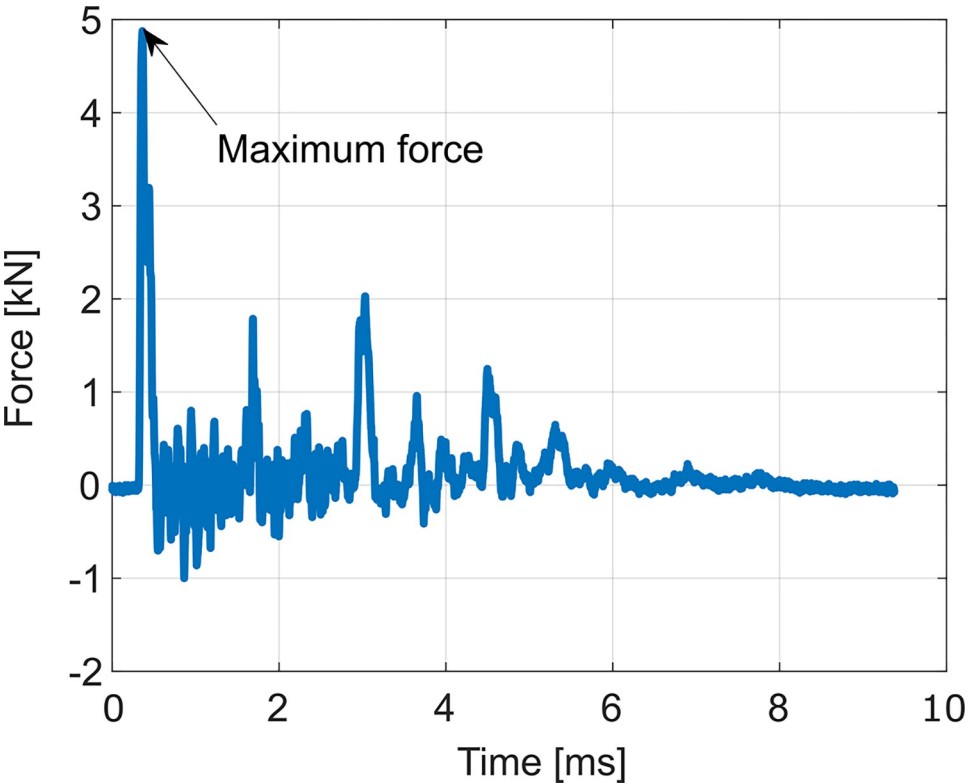

**Fig 4. Typical force signal for a single mallet stroke with the maximum force marked.**

[36]. A higher value for CCS is connected to femurs of Dorr type C (CCS < 1.18 Dorr Type A, [1.18, 1.54] Dorr Type B, CCS > 1.54 Dorr Type C) [36].

Each cavity preparation with the final broach and each stem implantation required a minimum of eight mallet strokes. The maximum peak force was calculated as the mean of the maxima of the last eight strokes applied. Fig 4 depicts a typical single mallet stroke and its maximum.

For the contact analysis the femur models were segmented from the CT-scans of the femurs with prepared cavities based on a threshold of 250 to 2000 mgHA/cm$^3$ to include trabecular as well as cortical bone (Fig 5A) [37–39]. The laser-scan of the femur with the implant was aligned to the CT-scan of the femur with the prepared cavity. The laser-scan of the implant was superimposed based on the proximal geometry from the scan of the femur with the implant (Fig 5A–5D). During data analysis the stem surface was subdivided into a proximal and a distal part based on the two different teeth surfaces of the Emphasys broaches (Fig 1A). As the Corail broaches do not show a similar change of the teeth, a corresponding Emphasys broach was superimposed on the Corail stem to identify a proximal part of comparable height. Contact analysis of the bone-implant interface was performed by calculating the percentage of bone with a gap less than 0.5 mm from the implant (Fig 5E). The actual press-fit was calculated as the mean of the distance of the interference overlap between bone and implant (Fig 5F). Contact and press-fit were analyzed for the total surface as well as the proximal and distal parts of the implant.

The final positions of the stem and the final broach in the femur cavity were compared using superimposed laser-scans to compare the seating behavior of the stems quantified by the angle between the broach and implant axis (seating rotation) and the height difference of stem

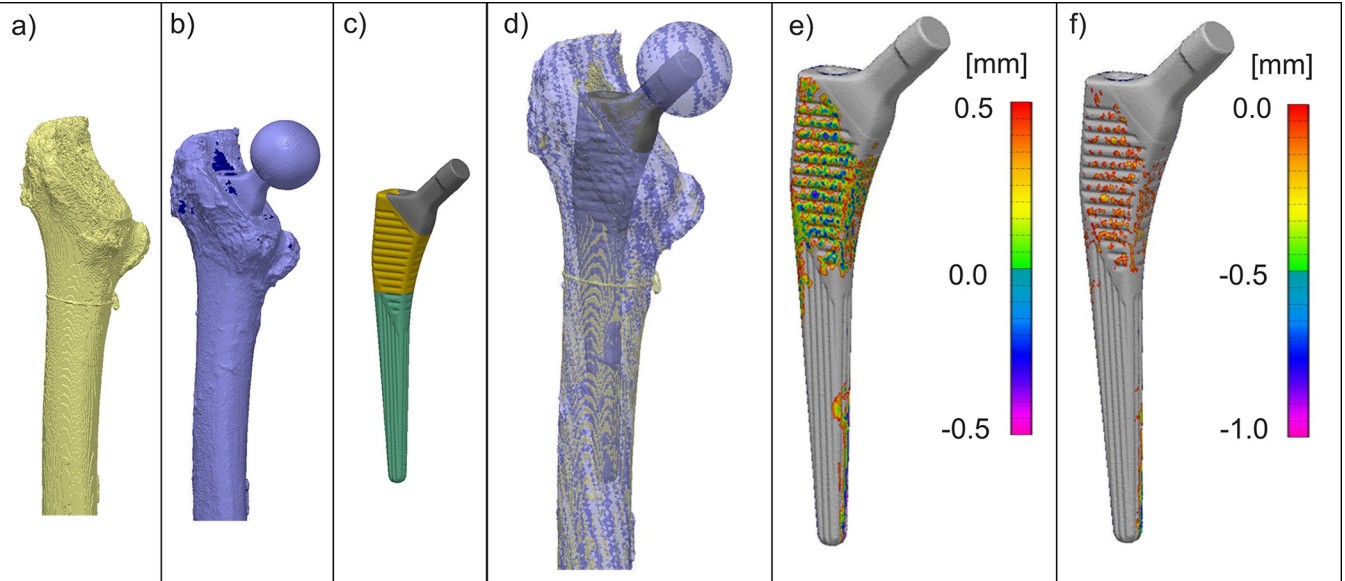

**Fig 5.** Workflow to determine the contact area and the actual press-fit in the bone-implant interface based on (a) a CT-scan of the prepared bone with cavity and rubber band tag to distinguish between femurs, (b) a laser-scan of the implanted stem and (c) a laser-scan of the stem (proximal part of the stem shown in yellow, distal part in green). (d) The superimposed combination of a-c. (e) Example for the contact area (in mm, positive values above 0.5 correspond to gaps; values below 0.5 are interpreted as contact), (f) example for the actual press-fit magnitude (in mm).

and broach (seating depth). The final position of the implant (varus / valgus tilt) and the canal-fill-ratio was also determined [36]. The canal-fill-ratio is the ratio between the stem area and the cavity area in a sagittal section of the implanted femur averaged along the entire stem [36].

Translational and rotational relative motions between femur and implant defined by respective markers assuming rigid body movement were determined during loading (GOM Correlate, GOM Software 2018, Carl Zeiss GOM Metrology GmbH, Braunschweig, Germany). The translational difference in position between the start of the first loading cycle and the end of the last loading cycle was defined as subsidence as was the rotational difference as total rotation (Fig 6).

## Statistical analysis

Statistical analyses were performed with a type I error level of 0.05 for all tests (SPSS 26.0, SPSS Inc., Chicago, NY, US). Paired statistical testing was performed after confirming the BMD distribution of the femur pairs using a non-paired t-test. Data was checked for normality and homogeneity of variance; parametric distributed data was analyzed using paired t-tests, non-parametric data using paired Wilcoxon tests. Pearson correlations were calculated for normally distributed data, Spearman's rho correlations otherwise. Two-way ANOVA was used when more than one factor was considered, and a mixed ANOVA was calculated for the repeated measurement of the translational and rotational relative motions. Outliers were kept in all analyses after checking for measurement errors. Type I error probabilities between 0.05 and 0.10 were denoted as trends. The statistical power was specified for test probabilities between 0.05 and 0.20.

## Results

The width of the Emphasys stem design is up to 4% (e.g. Size 10: Corail: 26.4 mm; Emphasys: 27.5 mm) and the thickness up to 7% larger (Size 10: Corail: 14.2 mm; Emphasys: 15.2 mm).

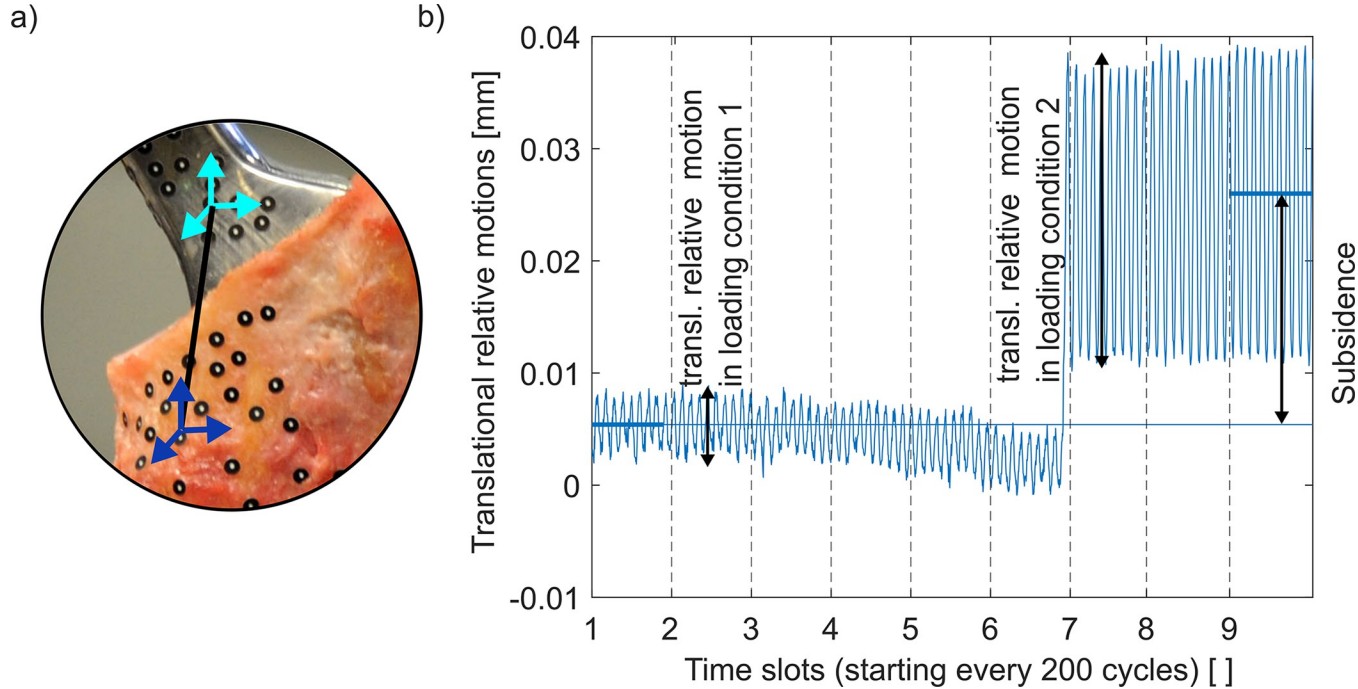

**Fig 6.** (a) Markers for the definition of the coordinate systems for the femur and the stem. (b) Example for the translational relative motion during the two loading conditions. The end of time slot 6 corresponds to the start of the high load and the associated increase in the translational relative motion.

The length is up to 7% shorter (Size 10: Corail: 140 mm; Emphasys: 132 mm) and the investigated cross-section is around 20% reduced for the Emphasys design (Size 10: Corail: 56 mm$^2$; Emphasys: 47 mm$^2$).

The BMD distribution on the right and left side of the femur pairs was similar (p = 0.6, Table 1), justifying the use of paired statistical tests. The BMD of the trabecular bone was similar for either stem design (Corail: 420.6 ± 41.1 mgHA/cm$^3$, Emphasys: 423.9 ± 53.5 mgHA/cm$^3$, p = 0.76). The CCS of the femurs used for the Corail stem was slightly smaller indicating more femurs of Dorr type A (Corail: 1.38 ± 0.34, Emphasys: 1.45 ± 0.36, p = 0.09, power = 0.0519, Table 1). Three out of six Corail stems and four out of six Emphasys stems were implanted one stem size below the templated size, one of the Emphasys stems two sizes below (Table 1).

### Broaching and implantation forces

The surgeon applied significantly higher broaching forces preparing the Emphasys cavities (Corail: 4.34 ± 1.28 kN; Emphasys: 6.09 ± 2.02 kN; p = 0.024). Impaction forces during implantation of the Emphasys stems also showed a statistical trend towards higher forces (Corail: 6.31 ± 1.15 kN, Emphasys: 7.10 ± 0.94 kN, p = 0.077, power: 0.1891). The number of strokes needed with either broach designs was similar (Corail: 17.2 ± 7.7, Emphasys: 20.8 ± 4.9, p = 0.329) as for both implant designs (Corail: 21.0 ± 5.3, Emphasys: 25.5 ± 7.6, p = 0.127). Impaction force did not increase with BMD during broaching (Corail: $R^2$ = 0.367, p = 0.202; Emphasys: $R^2$ = 0.240, p = 0.324) nor during stem implantation (Corail: $R^2$ = 0.561, p = 0.087; Emphasys: $R^2$ = 0.168, p = 0.420). The CCS ratio showed a negative correlation with force during broaching for either stem (Corail: $R^2$ = 0.797, p = 0.017; Emphasys: $R^2$ = 0.799, p = 0.016; Fig 7A). The direction was similar during implantation but the correlation was less

**Table 1. Specimen and implant details including the identification of the femur pair (PAIR), donor age (AGE), femur side (SIDE; L: Left, R: Right), bone mineral density (BMD), cortical-canal shape (CCS), bone morphology CCS converted to Dorr type (DORR), type of implant (STEM DESIGN), templated stem size (TEMP) and implanted stem size (IMP).** Since the implant size numbering scheme is different between the two designs, the corresponding Corail size is specified after the Emphasys size (in brackets).

| PAIR [] | AGE [YEARS] | SIDE [] | BMD [mgHA/cm$^3$] | CCS [] | DORR [] | STEM DESIGN | TEMP [] | INP [] |
|---|---|---|---|---|---|---|---|---|
| 1 | 65 | L | 449.7 | 1.28 | B | Corail | 13 | 13 |
| | | R | 435.9 | 1.34 | B | Emphasys | 8 or 9 (14 or 15) | 7 (13) |
| 2 | 45 | L | 329.7 | 2.04 | C | Emphasys | 5 (11) | 4 (10) |
| | | R | 342.9 | 1.84 | C | Corail | 10 | 10 |
| 3 | 64 | L | 412.8 | 1.69 | C | Corail | 12 | 12 |
| | | R | 430.2 | 1.63 | C | Emphasys | 7 (13) | 7 (13) |
| 4 | 59 | L | 494.2 | 1.46 | B | Emphasys | 6 (12) | 5 (11) |
| | | R | 453.4 | 1.37 | B | Corail | 12 | 11 |
| 5 | 58 | L | 423.0 | 1.16 | A | Corail | 11 | 10 |
| | | R | 438.4 | 1.23 | B | Emphasys | 6 (12) | 5 (11) |
| 6 | 37 | L | 415.0 | 0.99 | A | Emphasys | 6 (12) | 4 (10) |
| | | R | 441.5 | 0.93 | A | Corail | 11 | 10 |

pronounced for the Emphasys ($R^2$ = 0.767, p = 0.022, Fig 7B) and shown only as a trend for the Corail ($R^2$ = 0.539, p = 0.097, Fig 7B).

## Bone-to-implant interface

The bone-to-implant contact area over the entire implant was higher for the Emphasys stem (Corail: 47.0 ± 12.0%, Emphasys: 57.6 ± 12.0%, p = 0.004). A similar but non-significant relation was found for the proximal and the distal contact areas (Corail: proximal 57.6 ± 11.3%, distal 37.5 ± 16.5%; Emphasys: proximal 65.4 ± 8.1%, distal 50.1 ± 19.5%, p = 0.101). A higher

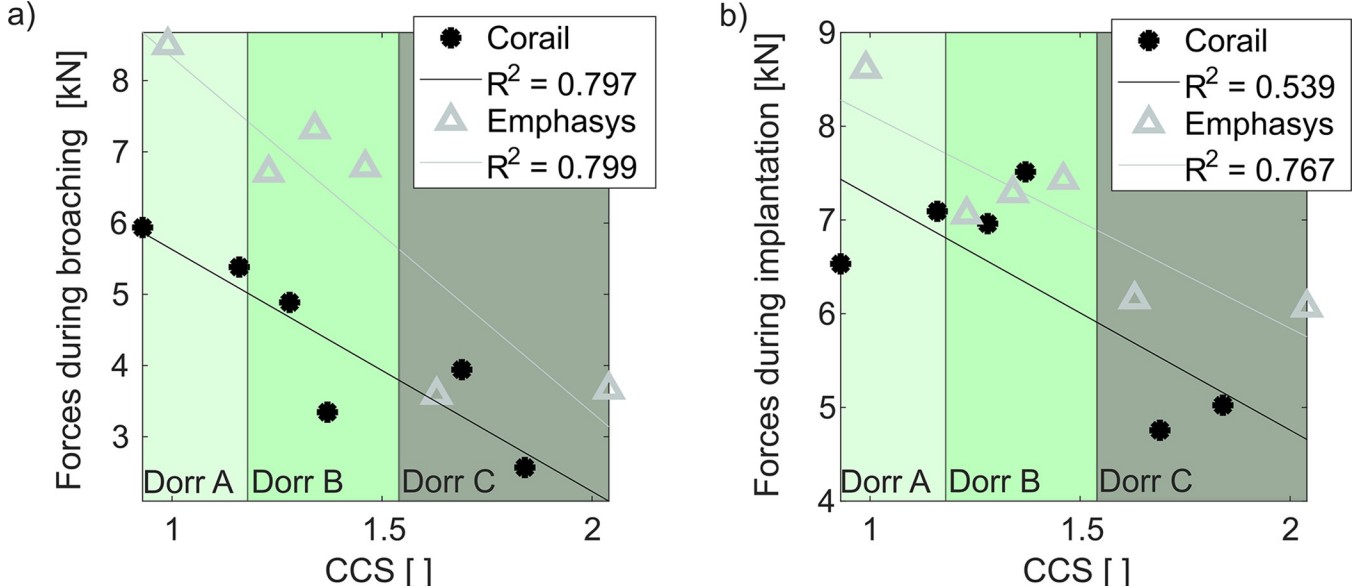

**Fig 7.** a) Forces during broaching decrease with increasing CCS ratio (Corail: $R^2$ = 0.797, p = 0.017; Emphasys: $R^2$ = 0.799, p = 0.016). (b) A similar effect but less pronounced was seen for stem implantation (Corail: $R^2$ = 0.539, p = 0.097; Emphasys: $R^2$ = 0.767, p = 0.022).

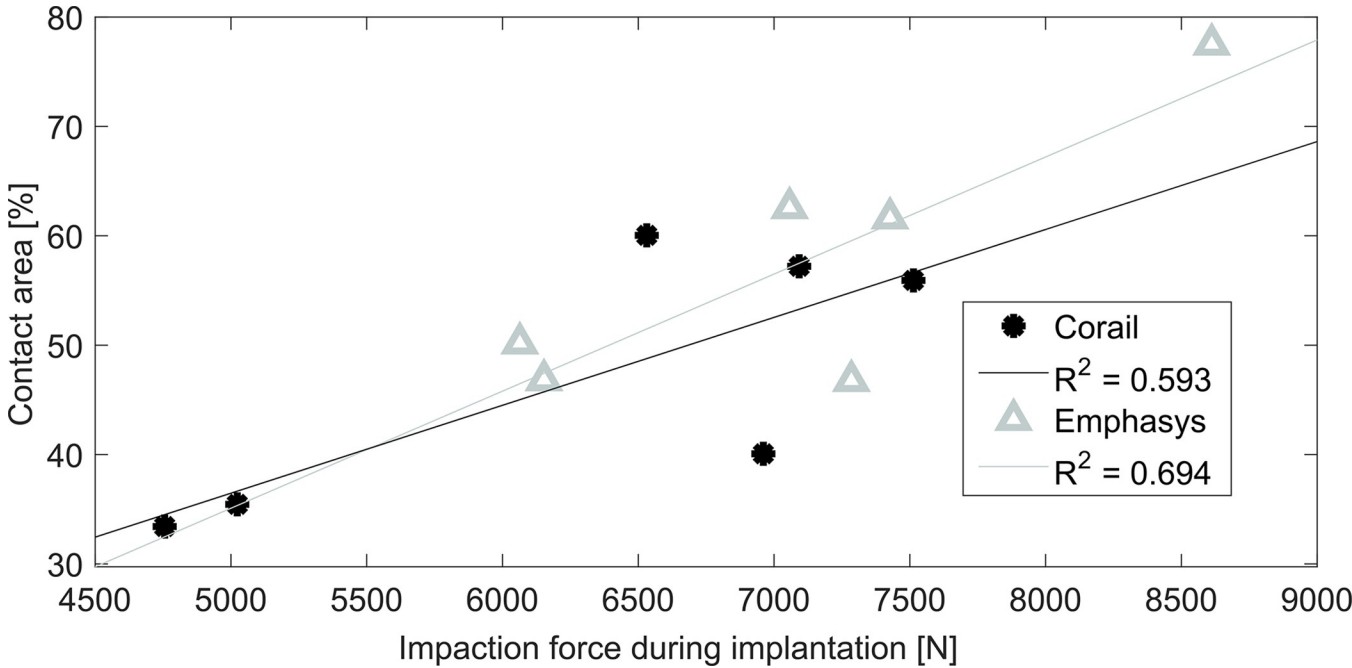

**Fig 8. Stem impaction force vs. total contact area for the two stem designs.** Higher contact areas were achieved for higher impaction forces for the Emphasys stem ($R^2$ = 0.694, p = 0.039). A trend towards a correlation was seen for the Corail ($R^2$ = 0.593, p = 0.073).

contact area was achieved proximally than distally for both stem designs (p = 0.007) and a higher contact area was achieved for higher implantation forces for both stem designs (Emphasys: $R^2$ = 0.694, p = 0.039; Corail: $R^2$ = 0.593, trend: p = 0.073; Fig 8).

Seating depth (Corail: 0.78 mm ($Q_1$ _ $Q_3$: 0.51 - 1.33 mm), Emphasys: 0.25 mm ($Q_1$ _ $Q_3$: 0.15 - 1.57 mm), p = 0.463, power: 0.088) and seating rotation (Corail: 1.7 ± 0.7˚, Emphasys: 1.8 ± 0.8˚, p = 0.604, power: 0.058) were similar for both stem designs.

The Corail stems showed a slight trend towards a valgus / negative inclination (- 0.88 ± 0.76˚, p = 0.109, power: 0.148), whereas the Emphasys stems tended to be closer to neutral (- 0.44 ± 0.78˚). The canal-fill-ratio was similar for either stem (Corail: 33.7 ± 6.4%, Emphasys: 34.8 ± 7.9%, p = 0.433, power: 0.057).

The higher nominal press-fit of the Emphasys stems (Corail: 0.414 ± 0.006 mm, Emphasys: 0.497 ± 0.035 mm, p = 0.006) was not reflected in the actual press-fit (Corail: 0.227 ± 0.034 mm; Emphasys: 0.224 ± 0.014 mm, p = 0.792).

## Relative motions due to loading

The subsidence after cyclic loading was lower for the Emphasys stem (Corail: 0.76 mm ($Q_1$ _ $Q_3$: 0.15 - 3.47 mm), Emphasys: 0.19 mm ($Q_1$ _ $Q_3$: 0.13 - 0.45 mm), p = 0.028, Fig 9A). The same as a trend was observed for the total rotation of the stem in the femur after cyclic loading (Corail: 1.74˚ ($Q_1$ _ $Q_3$: 0.32 - 4.41˚), Emphasys: 0.43˚ ($Q_1$ _ $Q_3$: 0.28 - 1.15˚), p = 0.075, power: 0.1414).

The stem design had no effect on the translational relative motions (p = 0.313). High loading significantly increased the translational relative motions (p = 0.011, Corail: low load: 0.025 ± 0.014 mm; high load: 0.104 mm ($Q_1$ _ $Q_3$: 0.088 - 0.121 mm); Emphasys: low load: 0.018 ± 0.007 mm, high load: 0.102 mm ($Q_1$ _ $Q_3$: 0.076 - 0.117 mm)). The magnitude of the increase was similar for either stem (p = 0.342).

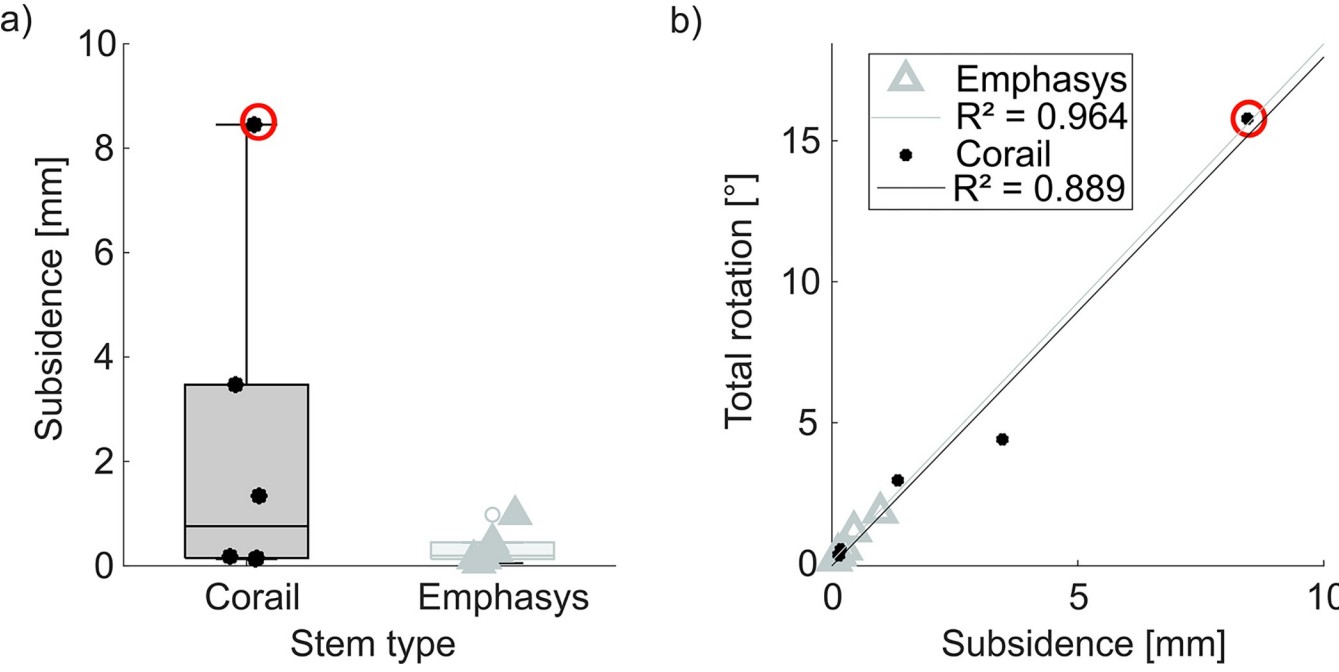

**Fig 9.** a) Subsidence after cyclic loading for both stem designs (whiskers indicate 1.5 times the IQR), red circles highlight the largest Corail value with subsidence over 8 mm; b) Total rotation increases with subsidence for both stem designs (Corail: $R^2 = 0.889$, p = 0.005; Emphasys: $R^2 = 0.964$, p < 0.001).

Higher loading also resulted in higher rotational relative motion (p = 0.002), while the stem design showed no effect (p = 0.105, low load: Corail: 0.14 ± 0.03˚, Emphasys: 0.10 ± 0.03˚; high load: Corail: 0.38˚ ($Q_1$ _ $Q_3$: 0.35 - 0.44˚), Emphasys: 0.30˚ ($Q_1$ _ $Q_3$: 0.29 - 0.32˚)).

Total rotation after cyclic loading increased with subsidence for both stem designs (Corail: $R^2 = 0.889$, p = 0.005; Emphasys: $R^2 = 0.964$, p < 0.001; Fig 9B).

The total contact area did not influence total rotation (Corail: $R^2 = 0.007$, p = 0. 872, Emphasys: $R^2 = 0.138$, p = 0.468), nor subsidence (Corail: $R^2 = 0.020$, p = 0.787, Emphasys: $R^2 = 0.138$, p = 0.468).

## Discussion

In this study a new stem design modified to reduce stem loosening especially in femurs of Dorr type A was compared to the original established design frequently used in cementless THA. The modified design showed reduced subsidence and total rotation after loading, indicating a higher primary stability which should clinically result in a reduced risk of loosening. Cavity broaching and stem implanting of the modified design required higher forces, which could increase the risk of PPFs. These results do not only apply to femurs of Dorr type A but to all Dorr categories, as in this study all Dorr types were used.

Most specimens had Dorr type A/B, as expected in middle-aged male patients under 70 years of age [40, 41], but one of the youngest patients, aged 45 years, showed femurs of Dorr type C (Table 1).

The results are limited by the small sample size and therefore low statistical power of this study due to the limited availability and high variability of human bone.

A distance of 70 mm between embedding and templated stem tip position was chosen independent from implant length and size to achieve a comparable bending situation [42]. The stem length varies between the stem designs as well as between the stem sizes. As a result, the

system stiffness is not identical for the different specimens, which might alter the reaction of the system to applied forces during cyclic loading.

The segmentation process of the CT-scans introduces a limitation, as the voxel length restricts the resolution of the surfaces to 0.4 mm. This introduces the possibility that the grey values of the CT-scans represent a mixture of trabecular or cortical bone with additional soft tissue. This has been addressed by setting 0.5 mm as a positive difference in the contact analysis.

The surgeon occasionally perceived higher forces required during preparation and stem implantation. As younger patients rarely undergo total hip arthroplasty, this was probably connected to the high bone quality of the study group. In a previous study the same surgeon applied up to 40% lower forces to femurs with 15% less BMD during preparation of the cavities [33]. However, higher BMD was not associated with higher impaction forces during preparation or implantation. One possible explanation would be that the forces were rather determined by the cortical bone and its morphology (Dorr Type and CCS) than by the density of the trabecular bone (BMD). Significant correlations of impaction forces during broaching and CCS were present. The higher forces in femurs of Dorr Type A can be explained by the larger amount of cortical bone that needs to be compacted or removed to prepare the cavities. In one of the Emphasys stems in a femur of Dorr type A, this was especially pronounced. Applying higher forces can be associated with an increased risk of PPFs [43], although none were seen in this study.

The implantation forces in Dorr Types A and B did not increase as much as during preparation. The correlations were strongly influenced by the femurs of Dorr type C, especially for the Corail stems. It is possible that other factors, such as the bone densification which was not investigated [17], might influence the impaction forces. Recording seating curves can give further information on the seating behavior which might be influenced by the component's geometry. This might affect the forces and will be investigated in further studies. The analysis of the stroke dynamics similar to [44, 45] might increase the understanding of the impaction process.

It is worth noting that when preparing the cavities for two of the Corail stems, the surgeon switched from a larger broach (not fully seated) back to one size smaller to avoid a potential PPF [5], in line with clinical practice. No stem failures were seen in these cases, suggesting that the larger broach had not yet noticeably widened the metaphyseal area. This indicates that this method can be acceptable if the broaching with a larger broach is stopped early enough. The differences observed between the two stem designs were not biased by those two cases in question.

The higher contact areas achieved with the Emphasys stems can be explained by the wider proximal design of the modified stem. Distally, the higher contact area was not expected for the Emphasys stems with their shorter and smaller distal geometry. As there was also no difference in the canal-fill-ratio, the higher contact could be explained by the more natural inclination of the stem, possibly changing the contact positions.

The better fixation of the Emphasys stems was probably due to the sharper edges of the new broaches (Fig 1A) [17, 46].

Distally the new broach design was developed to reduce the need for canal reaming. In the present study, the surgeon did not use canal reaming, so no comparison was possible.

The discrepancy between templated and implanted stem sizes does not primarily indicate undersized stems, but can be explained by surgeon preference, which has been shown to vary even among experienced surgeons [27]. This was particularly noticeable with the Emphasys stems, as the surgeon had not used the design before and therefore did not know how the stems would behave only based on the visual impression of the templates.

Nevertheless, complete stem seating was achieved for both stem designs, indicating a familiar procedure for the surgeon who subjectively reported no difference in the haptic feedback between the two impaction tips or stems. This is important as learning curves with surgical instruments and stems can affect outcomes and increase the number of PPFs [9, 47].

During cyclic loading most stems showed translational relative motions below 150 μm, low subsidence and low total rotation as intended for uncemented stems [1], as a prerequisite for osseointegration and long-term success of the implants. Considering, that the primary stability results for the Emphasys stem were at least similar to those of the Corail stems, a similar or even better performance of the Emphasys stem compared to the Corail stem can be expected.

The one Corail stem with more than 8 mm of subsidence was a typical example of a failed implantation with metaphyseal-diaphyseal mismatch in a femur of Dorr type A of an undersized stem (red circles in Fig 9) [11, 12, 48]. In the clinical situation such a large subsidence would probably have required a revision surgery. Clinical subsidence without consequence is reported for the range of 0.6 to 2.2 mm [29, 49]. Subsidence above 3 mm is viewed critically [50]. Whether the Emphasys stem of similar size implanted in the contralateral femur performed better due to the different design or its slightly higher CCS value (Corail: 1.16, Emphasys: 1.23) indicating a Dorr type B instead of a femur of Dorr type A, is not fully clear. The canal-fill-ratio was higher for the Emphasys stem in this femur pair (Corail: 0.286, Emphasys: 0.333). The improved canal-fill-ratio of the Emphasys stem could result in lower subsidence as well as a lower risk of loosening in the respective case. The risk of loosening for a potentially undersized stem should always be considered when implanting a stem smaller than the templated size [6, 7, 10–12]. Despite the fact that eight out of twelve specimens received stems one or two sizes smaller than the templated size of those eight, only one Corail stem exhibited severe subsidence which potentially would clinically lead to revision. The other seven stems sized smaller than planned would clinically not be regarded as undersized.

Future FE studies, based on the experiments, might provide additional insight whether the observed differences were solely stem driven or also partly dependent on the bone morphology.

With regard to the clinical performance of the collared version of the Corail stem [28, 51, 52] it can be expected that a collared Emphasys stem exhibits similar benefits.

## Conclusion

Minor design modifications of the fully porous HA-coated Corail tapered wedge stem, made with a focus on femurs of Dorr type A, resulted in an increased contact area as well as reduced subsidence and total rotation of the Emphasys stem after loading. Therefore, the risk of loosening is expected to be at least as low as for the established Corail stem [53]. The forces required to broach the femur and implant the new stem were higher–potentially increasing the risk of intraoperative PPF, which were not observed in this study. The shown benefits of the Emphasys stem have to be confirmed in the clinical setting or larger sized cohorts.

## Supporting information

**S1 Data. Raw study data containing all the data needed to recreate plots from the manuscript.**
(CSV)

## Author Contributions

**Conceptualization:** Tobias Konow.

**Data curation:** Katja Glismann, Tobias Konow.

**Formal analysis:** Katja Glismann.

**Funding acquisition:** Michael M. Morlock.

**Investigation:** Katja Glismann.

**Methodology:** Katja Glismann, Tobias Konow.

**Resources:** Frank Lampe, Benjamin Ondruschka.

**Software:** Katja Glismann.

**Supervision:** Tobias Konow, Gerd Huber, Michael M. Morlock.

**Visualization:** Katja Glismann.

**Writing – original draft:** Katja Glismann.

**Writing – review & editing:** Tobias Konow, Benjamin Ondruschka, Gerd Huber, Michael M. Morlock.

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
