## [Decision Letter · Decision Letter 0]

13 Dec 2023

PONE-D-23-33248Small design modifications can improve the primary stability of a fully coated tapered wedge hip stemPLOS ONE

Dear Dr. Glismann,

Thank you for submitting your manuscript to PLOS ONE. After careful consideration, we feel that it has merit but does not fully meet PLOS ONE’s publication criteria as it currently stands. Therefore, we invite you to submit a revised version of the manuscript that addresses the points raised during the review process.

We look forward to receiving your revised manuscript.

Kind regards,

Zhao Li, Ph.D., M.D.,

Academic Editor

PLOS ONE

“MMM is a paid consultant of DePuy Synthes and obtains research support as a Principal Investigator from Ceramtec, DePuy, and Beiersdorf. He obtains speaker’s fees from Aesculap, Ceramtec, DePuy, Zimmer, Peter Brehm, Corin, and Mathys andis in the editorial board “Trauma und Berufskrankheit.” GH is an associated member of the board of the German Society of Biomechanics.”

Reviewers' comments:

Reviewer's Responses to Questions

**Comments to the Author**

1. Is the manuscript technically sound, and do the data support the conclusions?

Reviewer #1: Partly

Reviewer #2: Yes

2. Has the statistical analysis been performed appropriately and rigorously? 

Reviewer #1: Yes

Reviewer #2: Yes

3. Have the authors made all data underlying the findings in their manuscript fully available?

Reviewer #1: Yes

Reviewer #2: Yes

4. Is the manuscript presented in an intelligible fashion and written in standard English?

Reviewer #1: Yes

Reviewer #2: Yes

5. Review Comments to the Author

Reviewer #1: 

The article presents a novel comparison of two stem designs, where one of them is modified to perform better in Dorr type A femurs. For that purpose, the modified stem was designed to be wider proximally. The authors also mentioned that the modified stem was slightly shorter and had a smaller tip, which is a relevant design modification for Dorr-type A femurs. The study covers all biomechanical aspects of initial stability, such as insertion forces, contact area, and post-operative mechanical testing. Therefore, in my opinion, the presented methodology and the results are valuable contributions to the current literature.

However, from my point of view, there are certain issues that need to be addressed. Overall, the introduction could establish a better connection with the previous literature. I understand the focus was only to compare two stem designs, but relating the current study with the previous experimental work focusing on the initial stem stability would be valuable. Likewise, discussion can also be improved, establishing a better connection with the related literature. The conclusion needs to be revised and should only focus on the main findings of the study. Apart from these, some specific issues in methods and results need to be addressed, as listed below.

I would like to categorize my suggestions as minor revisions because although some of the results might be influenced, the main results, the comparison between the two stem designs, should not change due to the suggested revisions. Mainly, I am concerned about the femur segmentation and the sensitivities that it might introduce. The authors provide a geometrical comparison under an mm tolerance. This could be substantially influenced by the segmentation settings, which need to be described and discussed in greater detail. Additionally, the authors did not provide any details regarding the evaluation of the hammering forces, which requires further clarification.

**Abstract:**

18-19: Implantation of an undersized stem is not the consequence of the “increased stem size increases the fracture risk.” Please consider revising the second sentence.

**Introduction:**

78-80: It is not clear what is meant; please reconsider revising the sentence.

80: Which workflow was referred to? It needs to be clarified; please revise.

87-89: The sentence is not clear; please consider revising it.

89-90: Cavity is an empty space, there will be no load distribution. Please revise the sentence.

94-96: The aim is clearly described, but this sentence is quite long. Please consider splitting it into more sentences.

**Methods:**

107-108: Are femurs cut in a way that they would have the same length? What was meant by “compatible”? Please clarify. If different femurs are cut into the same length, this would influence the biomechanics of the femur (maybe not substantial, but it still needs to be discussed in limitations).

153-155: Scan alignments require more explanation.

163-165: Not clearly described please revise.

166-170: Why were the intraoperative laser scans not used for the cavity surface? Was the scan quality for the cavity surface not good enough? Here, the cavity segmentation from the CT scan requires further detail because there might be differences in the segmented surface based on the segmentation settings. Since your analysis deals with geometrical differences of less than a millimeter, any geometrical deviation introduced during the CT segmentation might have a substantial influence on the results.

174-176: Why was the contact analysis conducted based on the bone-implant gap less than 0.5mm? If there is a positive gap between the surfaces, wouldn’t it mean no contact actually? I guess this sentence reads a bit misleading, what has been done became clear only by looking at the figure. Please consider rewriting it.

189: Apart from the corresponding citation, a brief explanation of the canal-fill ratio is necessary.

In methods, no specific information was given on how the broaching and insertion forces were evaluated. Hammering is a highly dynamic event, and assessing the applied forces during hammering is non-trivial. Therefore, this aspect requires further clarification. Presenting some of the raw force-time results would also be quite valuable. 

**Results:**

BMD values were evaluated in Ward’s triangles; it would also be valuable to report the t-score of the bone samples.

232-233: It is not clear in terms of what the observed “trend” was.

236-237: It would be nice to see the implantation forces vs BMD. It was said that the implantation forces did not change due to the BMD. Maybe not significantly, but looking at the p-value, there could still be a trend worth discussing. For example, based on a p-value of 0.329, it was written, “The number of strokes applied to the Emphasis broaches was numerically higher.” Therefore, one can also make a statement regarding the Force-BMD results based on a p-value of 0.202.

239-241: In terms of the insertion forces, the reported correlations disappear when the Dorr C samples are taken out. This needs to be discussed. 

265-267: It is not clear to me what is meant by “actually achieved press fit.”

Figure 7 shows the “contact area vs. Impaction forces during broaching”; however, in the text referring to figure 7, it was written “stem impaction forces.” Please clarify. Regarding the changes in the contact area, I would expect the stem insertion forces to be more relevant.

274-278: What is the difference between the “subsidence” and the “translational relative motion,” and why do we see differences between the two stem designs regarding subsidence and no differences in translational relative motion? Also, translational relative motion needs to be defined in methods.

288: “The total contact area was not influenced by the total rotation.” Was another scan conducted after the cyclic loading? How was this measured? Please clarify.

**Discussion:**

293-298: Here, a few more sentences would be helpful to remind the motivation and summary of what has been done. The second sentence of the discussion should not directly go into details of Dorr classification of individual femurs.

299-300: Limitations can be mentioned later in the discussion.

304-305: “Higher BMD within the study cohort in this study, however, was not associated with higher impaction forces.” This requires further explanation.

312: “No stem failures”: Should this be “no PPFs”?

312-314: It is unclear to me what is meant. Please clarify.

343-344: “The risk associated with undersizing must be emphasized in this context.” Please clarify the following: Risk of what? In which context?

344-345: “None of the other stems implanted one or two sizes below the templated size showed this behavior.” Please clarify: Which behavior?

The repetitive use of “this” in the final discussion paragraph makes it difficult to follow.

**Conclusion:**

349-350: The “speculation” mentioned here was not covered in the manuscript. Therefore, it does not belong to the conclusion.

351-352: “Increasing the risk of intraoperative PPF.” This is a hypothesis and should not be in conclusion.

The conclusion should only be written based on the main findings. Therefore, it needs to be revised.

Reviewer #2: Abstract:

• Clarify the number of femurs used in the study. You state "n = 6" but it's not clear if this refers to pairs or individual femurs.

Introduction Clarity:

The introduction provides a clear background and rationale for the study. It effectively communicates the problem, its relevance, and the objectives of the research.

• In the last sentence of the introduction, you mention "Dorr type A femurs," consider rephrasing as "femurs of Dorr type A."

Hypothesis and Objectives:

The hypothesis and objectives are well-stated. The study aims to address a practical problem in hip arthroplasty and evaluate the effectiveness of a modified stem design.

Literature Review:

The literature review is comprehensive and supports the need for the study. It effectively establishes the context and significance of the research.

Methodology:

• Clarify the number of femurs in each group. For example, in "Broaching and implantation forces for the modified stem were up to 40 % higher (p = 0.024)," mention how many femurs were used for this analysis.

• In "The surgeon was instructed to stop at a stem size below the preoperatively 120 planned size when he felt 121 that sufficient fixation of the stem was achieved and that an intraoperative fracture could occur 122 by increasing to the planned size," it would be beneficial to include the actual stopping criteria used by the surgeon.

The methodology is detailed and well-described, including the use of cadaveric femur pairs, the surgical procedures, and the testing conditions.

The statistical analysis plan is appropriately outlined, including the use of paired statistical tests and correlations.

Study Limitations:

The study acknowledges its limitations, such as the small sample size and potential variability in bone quality. This transparency enhances the credibility of the findings.

Results Presentation:

• Specify the units for the stem dimensions, broaching forces, and impaction forces (e.g., mm, kN).

• In Table 1, clarify the meaning of the columns with "PAIR," "AGE," "SIDE," "BMD," "CCS," "DORR," "DESIGN," "TEMPLATE," and "INSERT." It would be helpful to provide a legend or expand the abbreviations.

Results are presented clearly, with detailed information on stem dimensions, forces applied, contact areas, and other relevant parameters. Tables and figures effectively summarize the data.

Statistical Analysis:

The statistical analyses are appropriate for the study design. The use of paired tests for comparison between stem designs and correlation analyses adds robustness to the findings.

Discussion:

• Emphasize the clinical relevance and implications of the study findings. How might the observed changes in design influence real-world surgeries and patient outcomes?

• Discuss any potential limitations of the study, such as the small sample size, and suggest directions for future research.

The discussion interprets the results effectively and relates them to the study's objectives. The potential clinical implications of the findings are appropriately discussed.

The study's strengths and limitations are acknowledged in the discussion section.

Conclusion:

The conclusion is concise and summarizes the key findings. It emphasizes the potential benefits of the modified stem design in providing increased stability without increasing the risk of periprosthetic fractures.

Figures and Tables:

Figures and tables are used effectively to illustrate key points and present data. The information is well-organized and easy to follow.

Recommendations for Improvement:

• Consider providing additional information on the surgeon's experience and the training of the individuals involved in the surgical procedures.

• Include a discussion on the clinical relevance of the findings and potential implications for surgical practice.

Future Directions:

Consider discussing potential future research directions or clinical applications based on the study's findings.

Overall, the paper is well-structured, and the research is conducted with a rigorous methodology. The technical comments provided are meant to enhance the completeness and clarity of the manuscript.

General:

• Check the consistency of verb tenses throughout the paper. For example, in the methods section, you use past tense ("Initial CT-scans were taken..."), but some sentences are in present tense.

• Consider rephrasing complex sentences for better clarity.

References:

There are several numerical works are carried out in this field, the authors can refer thee papers and add in introduction part

• Wear estimation of hip implants with varying chamfer geometry at the trunnion junction: a finite element analysis

• Evolution of different designs and wear studies in total hip prosthesis using finite element analysis: A review

• Wear estimation at the contact surfaces of oval shaped hip implants using finite element analysis

• Optimization of Hip Implant Designs Based on Its Mechanical Behavior

• Static, dynamic, and fatigue life investigation of a hip prosthesis for walking gait using finite element analysis

• Finite element analysis of elliptical shaped stem profile of hip prosthesis using dynamic loading conditions

• Fatigue Life Evaluation of Different Hip Implant Designs Using Finite Element Analysis

Overall, the paper is well-structured, and the research is conducted with a rigorous methodology. The technical comments provided are meant to enhance the completeness and clarity of the manuscript.

6. PLOS authors have the option to publish the peer review history of their article (what does this mean?). If published, this will include your full peer review and any attached files.

Reviewer #1: No

Reviewer #2: No

---

## [Author Response · Author response to Decision Letter 0]

31 Jan 2024

RESPONSES TO THE REVIEWER COMMENTS

We would like to thank the editor and the reviewers for the efforts they have invested in the review and the helpful comments to improve the manuscript. We hope that our replies help to clarify any open questions. 

REVIEWER #1:

GENERAL COMMENT – PART A

The article presents a novel comparison of two stem designs, where one of them is modified to perform better in Dorr type A femurs. For that purpose, the modified stem was designed to be wider proximally. The authors also mentioned that the modified stem was slightly shorter and had a smaller tip, which is a relevant design modification for Dorr-type A femurs. The study covers all biomechanical aspects of initial stability, such as insertion forces, contact area, and post-operative mechanical testing. Therefore, in my opinion, the presented methodology and the results are valuable contributions to the current literature.

RESPONSE A

Thank you, we appreciate your positive feedback. Since most of the comments contain several points, we have split them up and addressed each of them separately. New text is highlighted in red while black text indicates parts of the original submission. 

General Comment – Part B

However, from my point of view, there are certain issues that need to be addressed. Overall, the introduction could establish a better connection with the previous literature. I understand the focus was only to compare two stem designs, but relating the current study with the previous experimental work focusing on the initial stem stability would be valuable. 

RESPONSE B

We see your point. Therefore, we added the following sentences to the introduction (lines 91 - 95): 

“A typical approach to analyze the consequences of these small design modifications would be a finite-element-analysis [19 – 23]. However, models always require an experimental validation by a biomechanical study. Experiments comparing different stem designs for primary stability [24,25] and micromotions [26-28] have been conducted in the past while subsidence is mostly analyzed from clinical studies [29-31].”

GENERAL COMMENT – PART C

Likewise, discussion can also be improved, establishing a better connection with the related literature. 

RESPONSE C

We added the following literature to the discussion section: 

Lines 378 - 380: “In the clinical situation such a large subsidence would probably have required a revision surgery. Clinical subsidence without consequence is reported for the range of 0.6 to 2.2 mm [30, 46]. Subsidence above 3 mm is viewed critically [47].”

Lines 394 - 395: “With regard to the clinical performance of the collared version of the Corail stem [29, 48, 49] it can be expected that a collared Emphasys stem exhibits similar benefits.”

GENERAL COMMENT – PART D

The conclusion needs to be revised and should only focus on the main findings of the study. Apart from these, some specific issues in methods and results need to be addressed, as listed below.

I would like to categorize my suggestions as minor revisions because although some of the results might be influenced, the main results, the comparison between the two stem designs, should not change due to the suggested revisions. Mainly, I am concerned about the femur segmentation and the sensitivities that it might introduce. The authors provide a geometrical comparison under an mm tolerance. This could be substantially influenced by the segmentation settings, which need to be described and discussed in greater detail. Additionally, the authors did not provide any details regarding the evaluation of the hammering forces, which requires further clarification.

RESPONSE D

Thank you, we appreciate your helpful feedback. We have addressed the question regarding segmentation in comment 10 and the analysis of the hammering forces in comments 9 and 13.

COMMENT 1 (ABSTRACT)

18-19: Implantation of an undersized stem is not the consequence of the “increased stem size increases the fracture risk.” Please consider revising the second sentence.

RESPONSE 1

Thank you for this comment. We agree and changed it (lines 18 - 20): 

“If in contrast a stem is implanted and sufficient primary stability is not achieved, such preventing successful osseointegration due to increased micromotions, it may also fail, especially if the stem is undersized.”

COMMENT 2 (INTRODUCTION)

78-80: It is not clear what is meant; please reconsider revising the sentence.

RESPONSE 2

We are sorry and changed the phrasing in lines 79 - 82: 

“Precise templating can reduce the amount of incorrect stem sizing and is an absolute necessity [15]. A well-defined workflow during surgery, including mandatory templating and adherence to the preoperative planned size during implantation, was shown to reduce the size-related revisions [9, 16].” 

COMMENT 3 (INTRODUCTION)

80: Which workflow was referred to? It needs to be clarified; please revise.

 RESPONSE 3

Thank you for this comment. We hope that the changes made due to comment 2 clarify the point. 

COMMENT 4 (INTRODUCTION)

87-89: The sentence is not clear; please consider revising it.

RESPONSE 4

We agree with you and changed it to the following sentence in lines 87 - 88: 

“Even for a distally fixed stem, the modified (larger) geometry of the implant in the metaphyseal area with a larger bone-implant interface should be beneficial [18].” 

COMMENT 5 (INTRODUCTION)

89-90: Cavity is an empty space, there will be no load distribution. Please revise the sentence.

RESPONSE 5

Thank you for clearing up. A more comprehensible version would be in lines 88 - 90: 

“The modifications might also alter the local bone loading due to the different wedging effect of the modified broaches and stems [14].”

COMMENT 6 (INTRODUCTION)

94-96: The aim is clearly described, but this sentence is quite long. Please consider splitting it into more sentences.

RESPONSE 6

Thank you for this comment, we shortened the sentences in lines 96 - 100: 

“The aim of the given study was [therfore] to investigate whether small design modifications of an established stem design can provide an improved primary stability without increasing the PPF risk. The design modifications include shortening of the stem and widening of the proximal stem as well as narrowing the stem at the tip. Those modifications are expected to be beneficial especially in femurs of Dorr type A.”

COMMENT 7 (METHODOLOGY)

107-108: Are femurs cut in a way that they would have the same length? What was meant by “compatible”? Please clarify. If different femurs are cut into the same length, this would influence the biomechanics of the femur (maybe not substantial, but it still needs to be discussed in limitations).

RESPONSE 7

Thank you for this input. In our study we worked with the medial and proximal part of the femurs. The distal part was removed to enable fixation. With “comparable lengths” we mean that the specimens were tailored depending on the templated stem size. We have included an additional sentence in lines 113 - 116: 

“The femurs were thawed at room temperature prior to testing [33]. The distal femur was sectioned 130 mm distally from the templated stem tip position and embedded in metal adapter flanges to a height of 60 mm (Technovit 4004; Kulzer, Germany). This created a consistent distance of 70 mm between embedding and the tip of the implant [34].”

We also added a paragraph to the discussion in lines 323 - 324: 

“A distance of 70 mm between embedding and templated stem tip position was chosen to achieve a comparable bending situation independent from implant length [42].”

COMMENT 8 (METHODOLOGY)

153-155: Scan alignments require more explanation.

RESPONSE 8

Thank you for this remark, we have added further information to clarify the alignment process (lines 165 - 170):

“The nominal press-fit was calculated as the mean distance between the stem surface and the corresponding broach surface (PolyWorks|Inspector 2020; InnovMetric Software, Canada). The respective geometries were derived from laser-scans. These were aligned directionally based on the axes of the distal sections of the stem and broach and positionally by aligning the top surface of the broach (attachment of the handle) with the proximal end of the implant coating (which it corresponds to).”

COMMENT 9 (METHODOLOGY)

163-165: Not clearly described please revise.

RESPONSE 9

We used “the maximum force that occurs during impaction” to serve as the parameter “peak force”. This single parameter is a simplification of the highly dynamic event that is happening and common practice [38,39]. We added the following information to clarify this (see lines 178 - 180):

“Each cavity preparation with the final broach and each stem implantation required a minimum of eight mallet strokes. The maximum peak force was calculated as the mean of the maxima of the last eight strokes applied [38, 39].” 

COMMENT 10 (METHODOLOGY)

166-170: Why were the intraoperative laser scans not used for the cavity surface? Was the scan quality for the cavity surface not good enough? Here, the cavity segmentation from the CT scan requires further detail because there might be differences in the segmented surface based on the segmentation settings. Since your analysis deals with geometrical differences of less than a millimeter, any geometrical deviation introduced during the CT segmentation might have a substantial influence on the results.

RESPONSE 10

Good point, this needs clarification, thank you for this comment. During the experiments the intraoperative laser scans of the femurs were obtained with the broach re. stem in situ and consequently we couldn’t obtain scans of the cavities. If we had tried scanning the femur cavity, we probably would have gotten insufficient points as the dimensions of the cavity entrance are rather small (around 400 mm²) and the angles needed to get proper triangulation would not work with our equipment. We could probably only record a maximum depth of 15 mm into the cavity. This would not include enough surface area for the contact analysis. Additionally, the scanner is affected by dark and wet surfaces which might introduce artefacts especially when cavities are recorded. 

We take your point that the segmented surface does have a large influence on the results. We therefore kept the threshold for segmentation constant. We have included this aspect in our limitations section in lines 325 - 328:

“The segmentation process of the CT-scans introduces a limitation, as the voxel length restricts the resolution of the surfaces to 0.4 mm. This introduces the possibility that the grey values of the CT-scans represent a mixture of trabecular or cortical bone with additional soft tissue. This has been addressed by setting 0.5 mm as a positive difference in the contact analysis.” 

COMMENT 11 (METHODOLOGY)

174-176: Why was the contact analysis conducted based on the bone-implant gap less than 0.5mm? If there is a positive gap between the surfaces, wouldn’t it mean no contact actually? I guess this sentence reads a bit misleading, what has been done became clear only by looking at the figure. Please consider rewriting it.

RESPONSE 11

Thank you for this comment, you are right, it would technically mean no contact. However, as you already stated in Comment 10, segmentation can highly affect the results of the contact analysis. We therefore wanted to omit the possible error of a wrongly segmented element (length 0.4 mm) in our contact analysis. In combination with possible alignment differences, we feel that 0.5 mm is an appropriate compromise taking the possible inaccuracy of the different steps of the analysis into account. 

Areas where the distance between the prosthesis and the bone is less than 0.5 mm are included in the contact area analysis. A distance larger than 0.5 mm is seen as no contact. The press fit magnitude shown in Figure 4 f) is determined only for areas with overlap between stem and bone (negative values).

We hope, this clarifies our method. We have adopted the figure caption (lines 197 - 199):

 “Figure 4: [..] (e) Example for the contact area (in mm, positive values above 0.5 correspond to gaps; values below 0.5 are interpreted as contact), (f) example for the actual press-fit magnitude (in mm).”

COMMENT 12 (METHODOLOGY)

189: Apart from the corresponding citation, a brief explanation of the canal-fill ratio is necessary.

RESPONSE 12

We appreciate this comment and have added the following information in lines 204 - 205: 

“The canal-fill-ratio is the ratio between the stem area and the cavity area in a sagittal section of the implanted femur averaged along the entire stem [37].”

COMMENT 13 (METHODOLOGY)

In methods, no specific information was given on how the broaching and insertion forces were evaluated. Hammering is a highly dynamic event, and assessing the applied forces during hammering is non-trivial. Therefore, this aspect requires further clarification. Presenting some of the raw force-time results would also be quite valuable. 

RESPONSE 13

Thank you for this comment. We have added the following information to the recording of the signal in lines 137 - 140: 

“Recording of the force data for a duration of 65 ms was triggered by an increase in the force signal above the threshold of 300 N (sample rate of 800 kHz, LabVIEW 2019, National Instruments, Austin, TX, US). A peak length of 0.3 ms for a single stroke was expected.” 

We hope this together with Response 9 explains our approach sufficiently. We would like to refrain from including the signal as a figure since we feel that it does not add any further information to images shown elsewhere [38, 39]. 

COMMENT 14 (RESULTS)

BMD values were evaluated in Ward’s triangles; it would also be valuable to report the t-score of the bone samples.

RESPONSE 14

We agree that the t-score would be valuable additional information. However, since we only receive the excised femurs, we do not have any information about the patients themselves and cannot take the soft tissue into account - which is needed for the t-score. The reported BMD only depends on the femurs itself and can also be reproduced with a calibrated CT (in vitro and in vivo). 

COMMENT 15 (RESULTS)

232-233: It is not clear in terms of what the observed “trend” was.

RESPONSE 15

Thank you for this comment. We call p values between 0.05 and 0.10 a “trend” (lines 225 - 226). We hopefully clarified the “trend” now in lines 251 - 253: 

“Impaction forces during implantation of the Emphasys stems also showed a statistical trend towards higher forces (Corail: 6.31 ± 1.15 kN, Emphasys: 7.10 ± 0.94 kN, p = 0.077, power: 0.1891).”

COMMENT 16 (RESULTS)

236-237: It would be nice to see the implantation forces vs BMD. It was said that the implantation forces did not change due to the BMD. Maybe not significantly, but looking at the p-value, there could still be a trend worth discussing. For example, based on a p-value of 0.329, it was written, “The number of strokes applied to the Emphasis broaches was numerically higher.” Therefore, one can also make a statement regarding the Force-BMD results based on a p-value of 0.202.

RESPONSE 16

Thank you for this response. We should not have used the expression “numerically higher” since the p-value does not confirm this, this was our mistake. Thank you for giving us the chance to correct it. We rephrased lines 253 - 256: 

“The number of strokes needed with either broach design was similar (Corail: 17.2 ± 7.7, Emphasys: 20.8 ± 4.9, p = 0.329) as for both stem designs (Corail: 21.0 ± 5.3, Emphasys: 25.5 ± 7.6, p = 0.127). “

For the second part of your comment, here are the detailed results of the analysis: 

“Impaction force did not increase with BMD during broaching (Corail: R2 = 0.367, p = 0.202; Emphasys: R2 = 0.240, p = 0.324) […]. “, (line 256)

The low force values for the lowest BMD in both stem designs can be explained by one specific femur pair with poor bone quality. Analysing the other five pairs alone shows clearly that there is no trend (Corail: R2 = 0.09, p = 0.624; Emphasys: R2 = 0.09, p = 0.624). 

“Impaction force did not increase with BMD […] during stem implantation (Corail: R2 = 0.561, p = 0.087; Emphasys: R2 = 0.168, p = 0.420).”, (line 257)

Again, the one pair with low BMD exhibits low forces but for the Corail with higher BMD even lower forces are measured. When left out again the use of the mean would be a good alternative to the regression analysis (Corail: R2 = 0.49, p = 0.188; Emphasys: R2 = 0.01, p = 0.873). 

This is why we did not put a larger focus on any trends connected between BMD and broaching or implantation forces. We will however, further discuss it in our Comment to Response 24 that concerns the discussion section. 

COMMENT 17 (RESULTS)

239-241: In terms of the insertion forces, the reported correlations disappear when the Dorr C samples are taken out. This needs to be discussed. 

RESPONSE 17

Thank you for this comment, especially in the context of our answer to Comment 16 you are right and this needs to be explained. We added the following sentences in the discussion in lines 336 - 345:

“Significant correlations of impaction forces during broaching and CCS were present. The higher forces in femurs of Dorr type A can be explained by the larger amount of cortical bone that needs to be compacted or removed to prepare the cavities. In one of the Emphasys stems in a femur of Dorr type A, this was especially pronounced. Applying higher forces can be associated with an increased risk of PPFs [43], although none were seen in this study. The implantation forces in Dorr Types A and B did not increase as much as during preparation. The correlations were strongly influenced by the femurs of Dorr type C, especially for the Corail stems. It is possible that other factors, such as the bone densification which was not investigated [18], might have influenced the impaction forces.”

COMMENT 18 (RESULTS)

265-267: It is not clear to me what is meant by “actually achieved press fit.”

RESPONSE 18

We are sorry for the confusion we might have caused by using a different name “actual achieved press-fit” in line 287 instead of the already introduced “Actual press-fit” from line 191-192: 

 “[…] was not reflected in the actual press-fit (Corail: 0.227 ± 0.034 mm; Emphasys: 0.224 ± 0.014 mm, p = 0.792).” (lines 285 - 286)

COMMENT 19 (RESULTS)

Figure 7 shows the “contact area vs. Impaction forces during broaching”; however, in the text referring to figure 7, it was written “stem impaction forces.” Please clarify. Regarding the changes in the contact area, I would expect the stem insertion forces to be more relevant.

RESPONSE 19

We are sorry for the mistake. We have now included the corrected figure with this text reference and description (lines 271 – 273 and lines 274 - 276):

“[…] and a higher contact area was achieved for higher implantation forces for both stem designs (Emphasys: R² = 0.694, p = 0.039; Corail: R² = 0.593, trend: p = 0.073; Fig 7).” 

“Figure 7: Stem impaction force vs. total contact area for the two stem designs. Higher contact areas were achieved for higher impaction forces for the Emphasys stem (R² = 0.694, p = 0.039). A trend towards a correlation was seen for the Corail (R² = 0.593, p = 0.073).”

COMMENT 20 (RESULTS)

274-278: What is the difference between the “subsidence” and the “translational relative motion,” and why do we see differences between the two stem designs regarding subsidence and no differences in translational relative motion? Also, translational relative motion needs to be defined in methods.

RESPONSE 20

Thank you for this comment. Subsidence and translational relative motion are two different mechanisms. The subsidence quantifies the positional change between the start and the end of the dynamic loading indicating how deep the stem was pushed into the cavity. It is permanent since the stem will not move upwards again. The translational relative motion describes the micromotion between stem and bone during each loading cycles. It is mainly reversible due to elastic deformation. The term micromotion is not used because it could be falsely seen as the movement of the stems itself – ignoring the deformation of the femur. 

The definition is given in the methods section in lines 206 - 211 and described in Figure 5 (lines 212 – 215). 

COMMENT 21 (RESULTS)

288: “The total contact area was not influenced by the total rotation.” Was another scan conducted after the cyclic loading? How was this measured? Please clarify.

RESPONSE 21

Thank you for this comment. We did a correlation analysis to see whether high contact comes along with low subsidence or low total rotation. The results indicate no correlation. No other scans were conducted. We have changed the wording (lines 307 - 309): 

“The total contact area did not influence total rotation (Corail: R2 = 0.007, p = 0. 872, Emphasys: R² = 0.138, p = 0.468), nor subsidence (Corail: R2 = 0.020, p = 0.787, Emphasys: R² = 0.138, p = 0.468).” 

COMMENT 22 (DISCUSSION)

293-298: Here, a few more sentences would be helpful to remind the motivation and summary of what has been done. The second sentence of the discussion should not directly go into details of Dorr classification of individual femurs.

RESPONSE 22

We followed your advice and added the following short summary before we go into the Dorr types (lines 311 - 317): 

“In this study a new stem design modified to reduce stem loosening especially in femurs of Dorr type A was compared to the original established design frequently used in cementless THA. The modified design showed reduced subsidence and total rotation after loading, indicating a higher primary stability which should clinically results in a reduced risk of loosening. Cavity broaching and stem implantation of the modified design required higher forces, which could increase the risk of PPFs. These results do not only apply for femurs of Dorr type A but for all Dorr categories, as in this study femurs of all Dorr types were used.”

COMMENT 23 (DISCUSSION)

299-300: Limitations can be mentioned later in the discussion.

RESPONSE 23

Thank you for this comment, we agree with you that normally the limitations should be located at the end of the discussion. For this study, we would like to keep them at the beginning as they address the high variability between bones, which is discussed throughout the discussion. This allowed us to reduce repetitions of this fact. 

COMMENT 24 (DISCUSSION)

304-305: “Higher BMD within the study cohort in this study, however, was not associated with higher impaction forces.” This requires further explanation.

RESPONSE 24

We value your input on the importance of a possible influence of BMD but refer to Comment 16. We could not find any such correlations. We have added the following aspect to the discussion in lines 333 - 336. Further changes to this paragraph were already introduced in Response 16: 

“However, higher BMD was not associated with higher impaction forces during preparation or implantation. One possible explanation would be that the forces were rather determined by the cortical bone and its morphology (Dorr Type and CCS) than by the density of the trabecular bone (BMD).”

COMMENT 25 (DISCUSSION)

312: “No stem failures”: Should this be “no PPFs”?

RESPONSE 25

Thank you for this comment, but in this case, we want to address a larger group of possible stem failures especially including loosening. We had the situation that a larger broach was shortly tested by the surgeon. The attempt was interrupted early and instead a smaller stem was implanted. We wondered whether the larger broach might already have widened the cavity and as a consequence would prevent a proximal fixation of the stem. With a smaller stem we rather feared loosening [34] and not PPFs, as we would have expected with a larger stem. Please also see the answer to your next comment. 

COMMENT 26 (DISCUSSION)

312-314: It is unclear to me what is meant. Please clarify.

RESPONSE 26

Thank you for this comment. In our in vitro study we observed the clinical practice of reverting to a smaller stem size after cavity broaching with a larger size was attempted but not completed. We expected this to have an effect on the primary stability of the stems, since the press-fit was expected to be smaller than for ordinarily prepared stems. However, we did not see any differences like higher micromotions. It such seems acceptable to revert to a smaller stem size before the completion of cavity preparation with a larger broach size. We made changes to the text to clarify this (lines 348 - 352): 

“No stem failures were seen in these cases, suggesting that the larger broach had not yet noticeably widened the metaphyseal area. This indicates that this method can be acceptable if the broaching with a larger broach is stopped early enough. The differences observed between the two stem designs were not biased by those two cases in question.”

COMMENT 27 (DISCUSSION)

343-344: “The risk associated with undersizing must be emphasized in this context.” Please clarify the following: Risk of what? In which context?

RESPONSE 27

Thank you for this comment. Some reports describe femurs of Dorr type A needing revision due to loosening [6,7,10-12], attributed to the implantation of an undersized stem associated with larger micromotions that might lead to loosening. The modified stem design is supposed to fit better into the proximal femur, which was confirmed by similar or even better stability in the present study. Most implanted stem sizes did not correspond to the templated size. We did not encounter an unsatisfying performance of the Emphasys stems but caution is called for (lines 385 - 388): 

“The improved canal-fill-ratio of the Emphasys stem could result in lower subsidence as well as a lower risk of loosening in the respective case. The risk of loosening for a potentially undersized stem should always be considered when implanting a stem smaller than the templated size [6, 7, 10 – 12].” 

COMMENT 28 (DISCUSSION)

344-345: “None of the other stems implanted one or two sizes below the templated size showed this behavior.” Please clarify: Which behavior?

RESPONSE 28

Thank you for this comment. With this we wanted to refer to the behaviour of all other smaller as templated stems in comparison to the single Corail stem with severe subsidence. We changed it to (lines 388 - 391):

“Despite the fact that eight out of twelve specimens received stems one or two sizes smaller than the templated size of those eight, only one Corail stem exhibited severe subsidence which potentially would clinically lead to revision. The other seven stems sized smaller than planned would clinically not be regarded as undersized.”

COMMENT 29 (DISCUSSION)

The repetitive use of “this” in the final discussion paragraph makes it difficult to follow.

RESPONSE 29

Thank you for your comment. Sorry.

a) We changed the first “this” to also include the speculation of a lower loosening rate (Comment 30; lines 371 – 373):

“During cyclic loading most stems showed translational relative motions below 150 μm, low subsidence and low total rotation as postulated for uncemented stems [1] as a prerequisite for osseointegration and long-term success of the implants.”

b) The next “this” is changed to (lines 376 – 380): 

“The one Corail stem with more than 8 mm of subsidence was a typical example of a failed implantation with metaphyseal-diaphyseal mismatch in a femur of Dorr type A of an undersized stem (red circles in Fig 8) [11 – 13]. In the clinical situation such a large subsidence would probably have required a revision surgery. Clinical subsidence without consequence is reported for the range of 0.6 to 2.2 mm [30, 46]. Subsidence above 3 mm is viewed critically [47].

c) The next “this” is kept to connect the sentence to the discussion points ahead but “femur pair” was added (lines 383 – 384).

“The canal-fill-ratio was higher for the Emphasys stem in this femur pair (Corail: 0.286, Emphasys: 0.333).” 

d) For the last “this” please see comment 27. We replaced it with “respective” case (lines 385 – 386).

“The improved canal-fill-ratio of the Emphasys stem could result in lower subsidence as well as a lower risk of loosening in the respective case.”

COMMENT 30 (CONCLUSION)

349-350: The “speculation” mentioned here was not covered in the manuscript. Therefore, it does not belong to the conclusion.

RESPONSE 30

We agree that we did not present a straight storyline that justifies this “speculation”. Since the second reviewer advised us to emphasize the clinical relevance of the study findings in his comment 14, we have now tried to find a compromise between both reviewers’ opinions. We have added the following lines to the discussion (lines 371 - 375). Please also see our responses to your comment 29. 

“During cyclic loading most stems showed translational relative motions below 150 µm, low subsidence and low total rotation as intended for uncemented stems [1], as a prerequisite for osseointegration and long-term success of the implants. Considering, that the primary stability results for the Emphasys stem were at least similar to those of the Corail stems, a similar or even better performance of the Emphasys stem compared to the Corail stem can be expected.”

We also changed the conclusion sentence to (lines 399 - 400): 

“Therefore, the risk of loosening is expected to be at least as low as for the established Corail stem [50].”

COMMENT 31 (CONCLUSION)

351-352: “Increasing the risk of intraoperative PPF.” This is a hypothesis and should not be in conclusion.

RESPONSE 31

We feel that considering the changes made (lines 340 – 341), this could underline that high forces are also not desirable during the implantation process. With this statement, we would also like to consider the compromise with the second reviewer to demonstrate the clinical significance of the results.

COMMENT 32 (CONCLUSION)

The conclusion should only be written based on the main findings. Therefore, it needs to be revised.

RESPONSE 32

According to comments 30 and 31, we added further aspects to the discussion so that the conclusion is now hopefully acceptable. We prefer to relate our findings to the main reasons for possible revisions, since this provides a more comprehensive view of why the directly tested characteristics are important in the context of the clinical long-term success of stems. Based on the additional reasoning in the discussion, we ask you to reconsider our conclusion. 

We thank you for your really thorough revision of our manuscript!  

REVIEWER #2

GENERAL COMMENT

Overall, the paper is well-structured, and the research is conducted with a rigorous methodology. The technical comments provided are meant to enhance the completeness and clarity of the manuscript.

RESPONSE

Thank you for this positive feedback to our manuscript and for conducting this review.

COMMENT 1 (ABSTRACT)

Clarify the number of femurs used in the study. You state "n = 6" but it's not clear if this refers to pairs or individual femurs.

RESPONSE 1

Thank you for the remark, we have changed the description in lines 30 - 31 to “(n = 6 pairs)”.

COMMENT 2 (INTRODUCTION)

The introduction provides a clear background and rationale for the study. It effectively communicates the problem, its relevance, and the objectives of the research.

RESPONSE 2

We are happy you have this impression of our introduction. 

COMMENT 3 (INTRODUCTION)

In the last sentence of the introduction, you mention "Dorr type A femurs," consider rephrasing as "femurs of Dorr type A."

RESPONSE 3

Thank you for this comment, we rephrased this description throughout the whole manuscript (see lines: 18, 22, 70, 100, 174, 176, 239, 311, 316, 319, 337, 339, 343 and 398)

COMMENT4 (HYPOTHESIS AND OBJECTIVES)

The hypothesis and objectives are well-stated. The study aims to address a practical problem in hip arthroplasty and evaluate the effectiveness of a modified stem design.

RESPONSE 4

Thank you for this positive comment. 

COMMENT 5 (INTRODUCTION)

The literature review is comprehensive and supports the need for the study. It effectively establishes the context and significance of the research.

RESPONSE 5

We thank you for this impression of our literature review. 

COMMENT 6 (METHODOLOGY)

Clarify the number of femurs in each group. For example, in "Broaching and implantation forces for the modified stem were up to 40 % higher (p = 0.024)," mention how many femurs were used for this analysis.

RESPONSE 6

Thank you for this response. We stated in lines 125 - 126 that six fresh frozen femoral pairs were used and added: 

“Six stems of each stem type were implanted according to the respective surgical instructions by an experienced surgeon [15, 35].”

COMMENT 7 (METHODOLOGY)

In "The surgeon was instructed to stop at a stem size below the preoperatively 120 planned size when he felt 121 that sufficient fixation of the stem was achieved and that an intraoperative fracture could occur 122 by increasing to the planned size," it would be beneficial to include the actual stopping criteria used by the surgeon.

RESPONSE 7

Thank you for your comment. The surgeon did not communicate a specific stop criterion and solely relied on his experience with a large number of operations with the same (Corail) or similar stem designs. In line 131 we have clarified that the surgeon decided based on his experience:

“He decided based on his experience.” 

COMMENT 8 (METHODOLOGY)

The methodology is detailed and well-described, including the use of cadaveric femur pairs, the surgical procedures, and the testing conditions. The statistical analysis plan is appropriately outlined, including the use of paired statistical tests and correlations.

RESPONSE 8

We are thankful for your opinion. 

COMMENT 9 (LIMITATIONS)

The study acknowledges its limitations, such as the small sample size and potential variability in bone quality. This transparency enhances the credibility of the findings.

RESPONSE 9

Thank you for this remark. 

COMMENT 10 (RESULTS)

Specify the units for the stem dimensions, broaching forces, and impaction forces (e.g., mm, kN).

RESPONSE 10

We are sorry, that we missed the units and added them in line 253: 

 “(Corail: 6.31 ± 1.15 kN, Emphasys: 7.10 ± 0.94 kN, p = 0.077, power: 0.1891)”

Please also see our response to your comment 11 where we also added the units. 

COMMENT 11 (RESULTS)

In Table 1, clarify the meaning of the columns with "PAIR," "AGE," "SIDE," "BMD," "CCS," "DORR," "DESIGN," "TEMPLATE," and "INSERT." It would be helpful to provide a legend or expand the abbreviations.

RESPONSE 11

Thank you for your comment, we have added further explanation to the table caption (lines 242 - 247):

“Table 1: Specimen and implant details including the identification of the femur pair (PAIR), donor age (AGE), femur side (SIDE; L: left, R: right), bone mineral density (BMD), cortical-canal shape (CCS), bone morphology CCS converted to Dorr type (DORR), type of implant (STEM DESIGN), templated stem size (TEMP) and implanted stem size (IMP). Since the implant size numbering scheme is different between the two designs, the corresponding Corail size is specified after the Emphasys size (in brackets).

COMMENT 12 (RESULTS)

Results are presented clearly, with detailed information on stem dimensions, forces applied, contact areas, and other relevant parameters. Tables and figures effectively summarize the data.

RESPONSE 12

We are thankful for your opinion on our results. 

COMMENT 13 (RESULTS)

The statistical analyses are appropriate for the study design. The use of paired tests for comparison between stem designs and correlation analyses adds robustness to the findings.

RESPONSE 13

We thank you for this impression of our statistical analysis. 

COMMENT 14 (DISCUSSION)

Emphasize the clinical relevance and implications of the study findings. How might the observed changes in design influence real-world surgeries and patient outcomes?

RESPONSE 14

Thank you for your comment. We addressed this topic in different parts of the discussion and conclusion, trying to find a compromise between your recommendations and those of reviewer 1. 

We refer to lines 313 – 314 and 371 - 375 were we described the higher primary stability that can be expected due to the lower relative motions and subsidence. We see better chances on osseointegration of the stem and expect a lower risk of loosening. 

We also discuss the relationship between the higher implantation forces in lines 314 – 317 and 340 – 341 the potentially higher periprosthetic fracture risk. 

During the preparation process we encountered the switch back from a larger to a smaller broach size. The surgeon called that a commonly used practice. We then analysed this further for its influence on the primary stability of a stem in lines 346 - 352. 

Additionally, we discussed the point that femurs of different Dorr types were tested and not only the target group of young male patients (see lines 316 - 320). 

We think that with those remarks we address the real-world surgeries. Further statements would bear the risk to over-interpret the achieved test data since the number of samples/implants is still small and in vitro studies always come along with high variations.

COMMENT 15 (DISCUSSION)

Discuss any potential limitations of the study, such as the small sample size, and suggest directions for future research.

RESPONSE 15

We appreciate this comment.

The limitation of the small sample size is addressed in terms of the low statistical power of this study in lines 321 - 322. Based on the limited availability we were not able to focus our results solely on the target group of the new stem but include all Dorr Types. We see a positive performance in all femurs and therefore this limitation might even be beneficial. It is described in lines 316 – 320. 

We address the problem arising from a differently prepared cavity (first larger broach then smaller stem) but this does not seem to affect the results. It is addressed in lines 346 - 352. 

For future research directions we would suggest the analysis of a collared stems as well as finite-element analysis which we address in our response to your Comment 21 and 24. 

COMMENT 16 (DISCUSSION)

The discussion interprets the results effectively and relates them to the study's objectives. The potential clinical implications of the findings are appropriately discussed. The study's strengths and limitations are acknowledged in the discussion section.

RESPONSE 16

Thank you for this positive comment. 

COMMENT 17 (CONCLUSION)

The conclusion is concise and summarizes the key findings. It emphasizes the potential benefits of the modified stem design in providing increased stability without increasing the risk of periprosthetic fractures.

RESPONSE 17

Thank you for this comment as well. 

COMMENT 18 (FIGURES)

Figures and tables are used effectively to illustrate key points and present data. The information is well-organized and easy to follow.

RESPONSE 18

Thank you for this positive reflection of our work. 

COMMENT 19 (RECOMMENDATIONS FOR IMPROVEMENT:)

Consider providing additional information on the surgeon's experience and the training of the individuals involved in the surgical procedures.

RESPONSE 19

Thank you for this comment, we now included additional information about the surgeon’s experience. Please, see lines 118 - 121. No other individuals were concerned with the surgical procedures. 

“Preoperative templating (Velys Surgery, DePuy Synthes, Warsaw, IN, US) and implantation were conducted by an experienced senior orthopedic surgeon with more than 1000 implantations using the established stem design.”

COMMENT 20 (RECOMMENDATIONS FOR IMPROVEMENT)

Include a discussion on the clinical relevance of the findings and potential implications for surgical practice.

RESPONSE 20

Thank you for this recommendation. We refer to our answer to your comment 14 where we showed clinical implications that can be derived from the study results with different examples. We would like to highlight the influence on potential revisions we made in the conclusion (loosening and periprosthetic fractures) in lines 399 - 400 and 400 - 402. 

We hope that we have found a good compromise between both reviewer’s views.

COMMENT 21 (FUTURE DIRECTIONS)

Consider discussing potential future research directions or clinical applications based on the study's findings.

RESPONSE 21

Thank you for this idea of a new direction. We added that the collared versions of the stems might have an additional positive effect, since the collared Corail stems have already shown a superior behaviour in different registries (lines 394 - 395): 

“With regard to the clinical performance of the collared version of the Corail stem [29, 48, 49], it can be expected that a collared Emphasys stem exhibits similar benefits.”

COMMENT 22 (GENERAL)

Check the consistency of verb tenses throughout the paper. For example, in the methods section, you use past tense ("Initial CT-scans were taken..."), but some sentences are in present tense.

RESPONSE 22

Thank you for this comment, we are sorry, that we did not pay enough attention to this but have checked it now throughout (see lines 226, 262, 358).

COMMENT 23 (GENERAL)

Consider rephrasing complex sentences for better clarity.

RESPONSE 23

We are sorry, we went through the text shortening most of the critical sentences (e.g. see lines 96 – 100, 184).

COMMENT 24 (REFERENCES)

There are several numerical works are carried out in this field, the authors can refer these papers and add in introduction part

1) Wear estimation of hip implants with varying chamfer geometry at the trunnion junction: a finite element analysis

2) Evolution of different designs and wear studies in total hip prosthesis using finite element analysis: A review

3) Wear estimation at the contact surfaces of oval shaped hip implants using finite element analysis

4) Optimization of Hip Implant Designs Based on Its Mechanical Behavior

5) Static, dynamic, and fatigue life investigation of a hip prosthesis for walking gait using finite element analysis

6) Finite element analysis of elliptical shaped stem profile of hip prosthesis using dynamic loading conditions

7) Fatigue Life Evaluation of Different Hip Implant Designs Using Finite Element Analysis

RESPONSE 24

Thanks for your suggestions. Those are very interesting paper. However, some are related to wear analysis and not to fatigue of the stem itself and the bone-implant interface like in our manuscript. We would prefer not to include the wear papers, but will keep them in mind when we will continue with FE analyses. The remainder papers (number 4, 6 and 7) were included (lines 91 - 92): 

“A typical approach to analyze the consequences of these small design modification would be a finite-element-analysis [19 – 23].” 

Additionally, the potential of further FE analyses is emphasized by adding the following to the discussion (lines 392 - 393): 

“Future FE studies, based on the presented experiments, might provide additional insight whether the observed differences were solely stem driven or also partly dependent on the bone morphology.”

We thank you for your detailed review of our manuscript.

FURTHER CHANGES:

We would like to add the following changes to our manuscript concerning additional citations and an adaption to Fig 1. There are no content changes in this section

The next 14 papers were added to establish a better connection with the previous literature in the introduction (Reviewer 1: comment B; Reviewer 2: Comment 24):

[3] K. Soballe, E. Hansen, H. Brockstedt-Rasmussen, and C. Bunger, “Hydroxyapatite coating converts fibrous tissue to bone around loaded implants,” J. Bone Joint Surg. Br., vol. 75-B, no. 2, pp. 270–278, Mar. 1993, doi: 10.1302/0301-620X.75B2.8444949.

[19] R. M. A. Al-Dirini, S. Martelli, D. Huff, J. Zhang, J. G. Clement, T. Besier, and M. Taylor, “Evaluating the primary stability of standard vs lateralised cementless femoral stems – A finite element study using a diverse patient cohort,” Clin. Biomech., vol. 59, pp. 101–109, Nov. 2018, doi: 10.1016/j.clinbiomech.2018.09.002.

[20] M. Reimeringer, N. Nuño, C. Desmarais-Trépanier, M. Lavigne, and P. A. Vendittoli, “The influence of uncemented femoral stem length and design on its primary stability: a finite element analysis,” Comput. Methods Biomech. Biomed. Engin., vol. 16, no. 11, pp. 1221–1231, Nov. 2013, doi: 10.1080/10255842.2012.662677.

[21] J. V. Corda, C. K N, S. Bhat N, S. Shetty, S. Shenoy B, and M. Zuber, “Finite element analysis of elliptical shaped stem profile of hip prosthesis using dynamic loading conditions,” Biomed. Phys. Eng. Express, vol. 9, no. 6, p. 065028, Nov. 2023, doi: 10.1088/2057-1976/acfe14.

[22] J. Corda, K. N. Chethan, S. Shenoy, S. Shetty, S. Bhat, and M. Zuber, “Fatigue life evaluation of different hip implant designs using finite element analysis,” J. Appl. Eng. Sci., vol. 21, no. 3, pp. 896–907, 2023, doi: 10.5937/jaes0-44094.

[23] H. Göktaş, E. Subaşi, M. Uzkut, M. Kara, H. Biçici, H. Shirazi, K. N. Chethan, and Ş. Mihçin, “Optimization of Hip Implant Designs Based on Its Mechanical Behaviour,” in Biomechanics in Medicine, Sport and Biology, vol. 328, A. Hadamus, S. Piszczatowski, M. Syczewska, and M. Błażkiewicz, Eds., in Lecture Notes in Networks and Systems, vol. 328. , Cham: Springer International Publishing, 2022, pp. 37–43. doi: 10.1007/978-3-030-86297-8_4.

[24] K. J. Fischer, D. R. Carter, and W. J. Maloney, “In vitro study of initial stability of a conical collared femoral component,” J. Arthroplasty, vol. 7, pp. 389–395, Jan. 1992, doi: 10.1016/S0883-5403(07)80029-5.

[25] G. Demey, C. Fary, S. Lustig, P. Neyret, and T. A. si Selmi, “Does a Collar Improve the Immediate Stability of Uncemented Femoral Hip Stems in Total Hip Arthroplasty? A Bilateral Comparative Cadaver Study,” J. Arthroplasty, vol. 26, no. 8, pp. 1549–1555, Dec. 2011, doi: 10.1016/j.arth.2011.03.030.

[26] S. R. Small, S. E. Hensley, P. L. Cook, R. A. Stevens, R. D. Rogge, J. B. Meding, and M. E. Berend, “Characterization of Femoral Component Initial Stability and Cortical Strain in a Reduced Stem-Length Design,” J. Arthroplasty, vol. 32, no. 2, pp. 601–609, Feb. 2017, doi: 10.1016/j.arth.2016.07.033.

[27] V. Malfroy Camine, H. A. Rüdiger, D. P. Pioletti, and A. Terrier, “Effect of a collar on subsidence and local micromotion of cementless femoral stems: in vitro comparative study based on micro-computerised tomography,” Int. Orthop., vol. 42, no. 1, pp. 49–57, Jan. 2018, doi: 10.1007/s00264-017-3524-0.

[28] T. Konow, J. Bätz, D. Beverland, T. Board, F. Lampe, K. Püschel, and M. M. Morlock, “Variability in Femoral Preparation and Implantation Between Surgeons Using Manual and Powered Impaction in Total Hip Arthroplasty,” Arthroplasty Today, vol. 14, pp. 14–21, Apr. 2022, doi: 10.1016/j.artd.2021.10.005.

[29] D. J. Simpson, B. J. L. Kendrick, M. Hughes, S. Glyn-Jones, H. S. Gill, G. F. Rushforth, and D. W. Murray, “The migration patterns of two versions of the Furlong cementless femoral stem: A RANDOMISED, CONTROLLED TRIAL USING RADIOSTEREOMETRIC ANALYSIS,” J. Bone Joint Surg. Br., vol. 92-B, no. 10, pp. 1356–1362, Oct. 2010, doi: 10.1302/0301-620X.92B10.24399.

[30] D. Campbell, G. Mercer, K. G. Nilsson, V. Wells, J. R. Field, and S. A. Callary, “Early migration characteristics of a hydroxyapatite-coated femoral stem: an RSA study,” Int. Orthop., vol. 35, no. 4, pp. 483–488, Apr. 2011, doi: 10.1007/s00264-009-0913-z.

[31] C. Ries, C. K. Boese, F. Dietrich, W. Miehlke, and C. Heisel, “Femoral stem subsidence in cementless total hip arthroplasty: a retrospective single-centre study,” Int. Orthop., vol. 43, no. 2, pp. 307–314, Feb. 2019, doi: 10.1007/s00264-018-4020-x.

The video cited previously as [22] is now published in form of a peer reviewed paper, therefore the citation was changed from: 

[22] T. Konow, “Influence of Stem Size and Position on Periprosthetic Fracture Risk and Primary Stability in Total Hip Arthroplasty,” presented at the International Society for Technology in Arthroplasty (ISTA), Periprosthetic fractures, Hawaii, US, Sep. 02, 2022. Accessed: Sep. 01, 2023. [Online]. Available: https://www.vumedi.com/channel/innovations-in-the-treatment-of-femoral-periprosthetic-fractures-devices-data-and-techniques/tab/innovations-in-the-treatment-of-femoral-periprosthetic-fractures-devices-data-and-techniques/video/influence-of-stem-size-and-position-on-periprosthetic-fracture-risk-and-primary-stability-in-tha/

To: 

[34] T. Konow, K. Glismann, F. Lampe, B. Ondruschka, M. M. Morlock, and G. Huber, “Stem size and stem alignment affects periprosthetic fracture risk and primary stability in cementless total hip arthroplasty,” J. Orthop. Res., p. jor.25729, Nov. 2023, doi: 10.1002/jor.25729.

From citation [36] we first copied the canal rasp but exchanged it with a picture of the one used in this study. Therefore, it is only referred to as similar and Fig. 1 was adapted including the new canal finder rasp. 

[36] Innomed, Inc., “INNOMED Orthopedic Instruments.” 2020. Accessed: May 17, 2023. [Online]. Available: https://fischermedical.dk/wp-content/uploads/Innomed_EN_HipInstruments_Apr2020.pdf

The next two papers were added as a reference for similar studies concerning mallet strokes (Reviewer 1: Comments 9 and 13): 

[38] T. Wendler, T. Prietzel, R. Möbius, J.-P. Fischer, A. Roth, and D. Zajonz, “Quantification of assembly forces during creation of head-neck taper junction considering soft tissue bearing: a biomechanical study,” Arthroplasty, vol. 3, no. 1, p. 20, Dec. 2021, doi: 10.1186/s42836-021-00075-7.

[39] N. E. Bishop, P. Wright, and M. Preutenborbeck, “A Parametric Analysis of Femoral Stem Impaction,” In Review, preprint, Sep. 2021. doi: 10.21203/rs.3.rs-869970/v1.

Wendler et al. was added to compare the cutting lengths of the femurs to another study (methods and discussion, Reviwer 1: comment 7)

[42] T. Wendler, B. Fischer, A. Brand, M. Weidling, J. Fakler, D. Zajonz, and G. Osterhoff, “Biomechanical testing of different fixation techniques for intraoperative proximal femur fractures: a technical note,” Int. Biomech., vol. 9, no. 1, pp. 27–32, Dec. 2022, doi: 10.1080/23335432.2022.2142159.

Papers 46 - 49 were added to establish a better connection with the previous literature in the discussion (Reviewer 1: comment B):

[46] D. Dammerer, P. Blum, D. Putzer, D. Krappinger, C. Pabinger, M. C. Liebensteiner, and M. Thaler, “Migration characteristics of the Corail hydroxyapatite-coated femoral stem—a retrospective clinical evaluation and migration measurement with EBRA,” Arch. Orthop. Trauma Surg., vol. 142, no. 3, pp. 517–524, Mar. 2022, doi: 10.1007/s00402-021-03926-9.

[47] M. Al-Najjim, U. Khattak, J. Sim, and I. Chambers, “Differences in subsidence rate between alternative designs of a commonly used uncemented femoral stem,” J. Orthop., vol. 13, no. 4, pp. 322–326, Dec. 2016, doi: 10.1016/j.jor.2016.06.026.

[48] J.-P. Vidalain, “Twenty-year results of the cementless Corail stem,” Int. Orthop., vol. 35, no. 2, pp. 189–194, Feb. 2011, doi: 10.1007/s00264-010-1117-2.

[49] F. Syed, A. Hussein, K. Katam, P. Saunders, S. K. Young, and M. Faisal, “Risk of subsidence and peri-prosthetic fractures using collared hydroxyapatite-coated stem for hip arthroplasty in the elderly,” HIP Int., vol. 28, no. 6, pp. 663–667, Nov. 2018, doi: 10.1177/1120700017754085.

---

## [Decision Letter · Decision Letter 1]

16 Feb 2024

PONE-D-23-33248R1Small design modifications can improve the primary stability of a fully coated tapered wedge hip stemPLOS ONE

Dear Dr. Glismann,

Thank you for submitting your manuscript to PLOS ONE. After careful consideration, we feel that it has merit but does not fully meet PLOS ONE’s publication criteria as it currently stands. Therefore, we invite you to submit a revised version of the manuscript that addresses the points raised during the review process.

We look forward to receiving your revised manuscript.

Kind regards,

Zhao Li, Ph.D., M.D.,

Academic Editor

PLOS ONE

Journal Requirements:

Reviewers' comments:

Reviewer's Responses to Questions

**Comments to the Author**

1. If the authors have adequately addressed your comments raised in a previous round of review and you feel that this manuscript is now acceptable for publication, you may indicate that here to bypass the “Comments to the Author” section, enter your conflict of interest statement in the “Confidential to Editor” section, and submit your "Accept" recommendation.

Reviewer #1: (No Response)

Reviewer #2: All comments have been addressed

2. Is the manuscript technically sound, and do the data support the conclusions?

Reviewer #1: Partly

Reviewer #2: Yes

3. Has the statistical analysis been performed appropriately and rigorously? 

Reviewer #1: N/A

Reviewer #2: Yes

4. Have the authors made all data underlying the findings in their manuscript fully available?

Reviewer #1: Yes

Reviewer #2: Yes

5. Is the manuscript presented in an intelligible fashion and written in standard English?

Reviewer #1: Yes

Reviewer #2: Yes

6. Review Comments to the Author

Reviewer #1: Dear authors,

Thank you very much for your responses and for taking the time to edit the paper as well as discuss the comments in detail.

I don’t have anything else to add to most of the points. Please find below further comments that require further clarification and discussion.

Response 7:

Thank you for your response. It is way more transparent now. Nevertheless, I think the following statement needs to be still discussed in more detail, including potential limitations:

“A distance of 70 mm between embedding and templated stem tip position was chosen to achieve a

comparable bending situation independent from implant length [42].”

In the case of larger femurs, one should observe a larger stem tip to the distal femur, increasing the bending distance. This might have consequences regarding the stiffness of the whole system, which won’t be visible in this study conducted with a standardized bending distance. Therefore, I think these consequences and their impact on the results of this study need to be discussed in greater depth.

Response 9:

Thank you for your response. Although the average maximum peak force might be relevant to forces transferred to the femur (or deformation energy of the femur), it is also influenced by other factors, such as the impaction speed or the dynamic motion of the stem. For example, It is possible to see high initial peak values in both cases where the stem undergoes rather a large dynamic movement and does not deform the femur or stay in its position (final strokes) and transfer all the loading to the femur contribution more significantly to initial stability or the fracture risk. From this perspective, the choice of the averaged maximum peak force should be discussed in greater detail. For example, some other similar studies are using integrated force values for similar types of evaluations:

https://doi.org/10.1016/j.clinbiomech.2020.105006

https://doi.org/10.1016/j.jmbbm.2019.103535

It would add value to the study if a metric/parameter/criteria could be used that is directly related to the energy absorbed by the femoral deformation.

Response 10:

Thanks for your response and extending the discussion. However, if the segmentation plays an important role, then the HU threshold used during the segmentation needs justification.

Do you have a reference for the used values (not visible in the current version)? If not, I recommend including sensitivity analysis to quantify any possible influence on the main conclusions (i.e., given the low taper angle, small deviations in geometry (+- 0.4 mm) should yield large differences in the force-displacement response of the stem during the insertion).

Response 13:

Thank you for your response. Regarding the “including the signal as a figure since we feel that it does not add any further information to images shown elsewhere [38, 39].” When we look at the references, one of them reports ex-vivo results for ceramic head insertion, and the other one is a parametric study analyzing possible sensitivities of the hammering and stem insertion process based on an analytical approach. So, none of these studies cover the hammering force processing of an ex-vivo stem insertion. Therefore, I believe showing one example of the raw data and how the forces were analyzed would help the reader to understand and make scientific judgments based on your results.

Additional comment:

In figure 4, it can be seen a cerclage, probably for intraoperative fracture mitigation, was applied. This was not mentioned in the manuscript. A cerclage prevents fracture because it provides circumferential support, which also influences the circumferential stiffness. These changes might influence the insertion force measurements and need to be discussed.

Reviewer #2: Small design modifications can significantly enhance the primary stability of a fully coated tapered wedge hip stem. Through meticulous adjustments in geometry or material composition, the stem can achieve optimal fixation within the bone, reducing the risk of implant loosening or migration. Enhancements may include refining surface texture to promote osseointegration, optimizing stem length or diameter for better fit, or incorporating features to distribute load more evenly. Such modifications bolster initial stability, crucial for successful long-term implant function and patient mobility. Even seemingly minor alterations can yield substantial improvements, underscoring the importance of precision in orthopedic implant design.

Good work

7. PLOS authors have the option to publish the peer review history of their article (what does this mean?). If published, this will include your full peer review and any attached files.

Reviewer #1: No

Reviewer #2: No

---

## [Author Response · Author response to Decision Letter 1]

3 Mar 2024

RESPONSES TO THE REVIEWER COMMENTS #2

We would like to thank the editor and the reviewers for the efforts they have invested in the second round of the review and the helpful comments to improve the manuscript. We hope that our replies help to clarify any open questions. 

REVIEWER #1:

GENERAL COMMENT – PART A

Dear authors,

Thank you very much for your responses and for taking the time to edit the paper as well as discuss the comments in detail. I don’t have anything else to add to most of the points. Please find below further comments that require further clarification and discussion.

RESPONSE A

Thank you for the follow up questions. We again split the comments up and addressed each of them separately. New text is highlighted in red while black text indicates parts of the original submission.

RESPONSE 7:

Thank you for your response. It is way more transparent now. Nevertheless, I think the following statement needs to be still discussed in more detail, including potential limitations:

“A distance of 70 mm between embedding and templated stem tip position was chosen to achieve a

comparable bending situation independent from implant length [42].”

In the case of larger femurs, one should observe a larger stem tip to the distal femur, increasing the bending distance. This might have consequences regarding the stiffness of the whole system, which won’t be visible in this study conducted with a standardized bending distance. Therefore, I think these consequences and their impact on the results of this study need to be discussed in greater depth.

RESPONSE 7:

Thank you for your Response. We agree with you, this is always an issue with bone specimens. We have added an additional discussion to lines 327 - 331:

“A distance of 70 mm between embedding and templated stem tip position was chosen independent from implant length and size to achieve a comparable bending situation [43]. The stem length varies between the stem designs as well as between the stem sizes. As a result, the system stiffness is not identical for the different specimens, which might alter the reaction of the system to applied forces during cyclic loading.”

RESPONSE 9:

Thank you for your response. Although the average maximum peak force might be relevant to forces transferred to the femur (or deformation energy of the femur), it is also influenced by other factors, such as the impaction speed or the dynamic motion of the stem. For example, It is possible to see high initial peak values in both cases where the stem undergoes rather a large dynamic movement and does not deform the femur or stay in its position (final strokes) and transfer all the loading to the femur contribution more significantly to initial stability or the fracture risk. From this perspective, the choice of the averaged maximum peak force should be discussed in greater detail. For example, some other similar studies are using integrated force values for similar types of evaluations:

[I] https://doi.org/10.1016/j.clinbiomech.2020.105006

[II] https://doi.org/10.1016/j.jmbbm.2019.103535

It would add value to the study if a metric/parameter/criteria could be used that is directly related to the energy absorbed by the femoral deformation.

RESPONSE 9:

Thank you for this further clarification. Impaction with a hard-on-hard contact as in orthopedics is a complicated situation. We can see your point of giving more information on the impaction process as it can vary not only between specimens but also over the implantation process of a single stem. 

We will include the following figure in our manuscript to give an example for a typical signal (line 182): 

“Figure 4: Typical force signal for a single mallet stroke with the maximum force marked.”

The following sentence will reference to it (lines 179-181): 

“Each cavity preparation with the final broach and each stem implantation required a minimum of eight mallet strokes. The maximum peak force was calculated as the mean of the maxima of the last eight strokes applied. Figure 4 depicts a typical single mallet stroke and its maximum [38,39].“

Based on your idea, we have conducted an analysis of the integrated force values. The method was adopted from the methodology introduced by the papers you referenced as well as their sources ([III] https://doi.org/10.1371/journal.pone.0166778, [IV] https://doi.org/10.1115/1.4029505

and [V] https://doi.org/10.1177/095441191455243). 

The force signal is integrated from the time point when it first exceeds 300 N. The global maximum of a single stroke is included and the integration ends when the force signal falls below 300 N. The threshold of 300 N was taken to clearly identify the signal against the noise. Afterwards the signal was divided by the timespan of the integration to find an averagely applied force per stroke. This new parameter is called impaction momentum (see I to II). We again calculate the mean for the last eight strokes. 

During broaching, the impaction momentum was similar for both designs (Corail: 2198.98 N (Q1 – Q3: 1968.2 – 2585.1 N), Emphasys: 2652.22 N (Q1 – Q3: 2066.8 – 3561.4 N), p = 0.116). During implantation, however, the impaction momentum was higher for the Emphasys stems (Corail: 3265.57 N (Q1 – Q3: 2575.7 – 3603.8 N), Emphasys: 3916.86 N (Q1 – Q3: 3852.5 - 4581.6 N), p = 0.046). 

Figure 1: Impaction momentum during (a) broaching and (b) implantation for both stem designs. 

When now the results of the maximum peak forces during broaching are compared to the newly calculated impaction momentum (fig. 1 a) the significantly higher peak forces are only trend wise shown in the impaction momentum. This result is mirrored for the impaction momentum during implantation (fig. 1 b) were now the impaction momentum is showing a significant difference while the maximum peak force is only showing a trend towards higher values for the Emphasys stem. There is additionally a high correlation between the maximum peak forces during broaching and the impaction momentum during broaching (Corail: R² = 0.889; p = 0.005; Emphasys: R² = 0.785; p = 0.019) with significant results for the implantation, too (Emphasys: R² = 0.889; p = 0.005). Only the results during implantation for the Corail (R² = 0.236; p = 0.236) were again influenced by the two Dorr type C femurs.

With these results we conclude that calculating the impaction momentum yields more information on the stroke dynamics. In this specific study the trends of the results however, are similar: the peak forces as well as the impaction momentums tend to be higher for the Emphasys stems. We will keep in mind for further studies to include more of the stroke dynamics. Especially including seating curves will be one of the goals for upcoming studies and we thank you for the idea to include it. We hope you can accept our point of view. For further clarification in our manuscript we have added the following sentence to the discussion (lines 351 - 355): 

“It is possible that other factors, such as the bone densification which was not investigated [18], might influence the impaction forces. Recording seating curves can give further information on the seating behavior which might be influenced by the component’s geometry. This might affect the forces and will be investigated in further studies. The analysis of the stroke dynamics similar to [45,46] might increase the understanding of the impaction process.“ 

45. Dubory A, Rosi G, Tijou A, Lomami HA, Flouzat-Lachaniette CH, Haïat G. A cadaveric validation of a method based on impact analysis to monitor the femoral stem insertion. J Mech Behav Biomed Mater. 2020; 103:103535. 

46. Albini Lomami H, Damour C, Rosi G, Poudrel AS, Dubory A, Flouzat-Lachaniette CH, Haiat G. Ex vivo estimation of cementless femoral stem stability using an instrumented hammer. Clin Biomech. 2020; 76:105006. 

RESPONSE 10:

Thanks for your response and extending the discussion. However, if the segmentation plays an important role, then the HU threshold used during the segmentation needs justification.

Do you have a reference for the used values (not visible in the current version)? If not, I recommend including sensitivity analysis to quantify any possible influence on the main conclusions (i.e., given the low taper angle, small deviations in geometry (+- 0.4 mm) should yield large differences in the force-displacement response of the stem during the insertion).

RESPONSE 10:

Thank you for your Response. In Kluess et al. it is stated that bony material can be identified from HU values starting at 200 – 250 HU [39]. In combination with Garg et al. who identified a value of 275 HU to be the threshold [38], a value of 250 HU was chosen similar to Gargiulo et al. [40] and applied during the segmentation. 

We will add the following references for the HU threshold (lines 183 -185):

“For the contact analysis the femur models were segmented from the CT-scans of the femurs with prepared cavities based on a threshold of 250 to 2000 mgHA/cm³ to include trabecular as well as cortical bone (Fig 5a) [38-40].”

38. Garg A, Deland J, Walker PS. Design of Intramedullary Femoral Stems Using Computer Graphics. Eng Med. 1985;14(2):89–93. 

39. Kluess D, Souffrant R, Mittelmeier W, Wree A, Schmitz KP, Bader R. A convenient approach for finite-element-analyses of orthopaedic implants in bone contact: Modeling and experimental validation. Comput Methods Programs Biomed. 2009; 95(1):23–30. 

40. Gargiulo P, Pétursson T, Magnússon B, Bifulco P, Cesarelli M, Izzo GM, Magnúsdóttir G, Halldórsson G, Ludvigsdóttir GK, Tribel J, Jónsson H. Assessment of Total Hip Arthroplasty by Means of C omputed T omography 3 D Models and Fracture Risk Evaluation. Artif Organs. 2013; 37(6):567–73. 

We also appreciate the second aspect of your response and would like to refer to lines 363 - 364 where we addressed the different shapes of the stems in connection to the contact area. However, the forces were more affected by the canal shape during broaching. We are sorry, but we cannot make any statements about the displacement during implantation as this was not recorded (please see our answer to Response 9). We will keep it in mind for further experiments. 

“The higher contact areas achieved with the Emphasys stems can be explained by the wider proximal design of the modified stem.”

RESPONSE 13:

Thank you for your response. Regarding the “including the signal as a figure since we feel that it does not add any further information to images shown elsewhere [38, 39].” When we look at the references, one of them reports ex-vivo results for ceramic head insertion, and the other one is a parametric study analyzing possible sensitivities of the hammering and stem insertion process based on an analytical approach. So, none of these studies cover the hammering force processing of an ex-vivo stem insertion. Therefore, I believe showing one example of the raw data and how the forces were analyzed would help the reader to understand and make scientific judgments based on your results.

RESPONSE 13:

Thank you for your response. You are right and the studies we referenced to do not show signals that were recorded in a similar set-up or with a similar scope. Therefore, we will include a figure of our signal to improve the understanding and possibility to judge for the reader. Please see our answer to your response 9. 

ADDITIONAL COMMENT:

In figure 4, it can be seen a cerclage, probably for intraoperative fracture mitigation, was applied. This was not mentioned in the manuscript. A cerclage prevents fracture because it provides circumferential support, which also influences the circumferential stiffness. These changes might influence the insertion force measurements and need to be discussed.

RESPONSE

Thank you for this comment, we are sorry, for the confusion we might have caused with this but we did not use any cerclage during the experiments. In figure 4 a rubber band can be seen which was used to attache tags identifying the femurs during the CT-scans. We moved these rubber bands distally during the experiments (see figure 3). They were not used for any stabilization of the femur and could not change the results. However, we added the following sentence to the description (lines 197- 199): 

Figure 5: […] (a) a CT-scan of the prepared bone with cavity and rubber band tag to distinguish between femurs, (b) […]

We thank you for your really thorough revision of our manuscript!  

REVIEWER #2

GENERAL COMMENT

Small design modifications can significantly enhance the primary stability of a fully coated tapered wedge hip stem. Through meticulous adjustments in geometry or material composition, the stem can achieve optimal fixation within the bone, reducing the risk of implant loosening or migration. Enhancements may include refining surface texture to promote osseointegration, optimizing stem length or diameter for better fit, or incorporating features to distribute load more evenly. Such modifications bolster initial stability, crucial for successful long-term implant function and patient mobility. Even seemingly minor alterations can yield substantial improvements, underscoring the importance of precision in orthopedic implant design.

Good work

RESPONSE

Thank you again for conducting the review to our manuscript and the positive feedback.

FURTHER CHANGES:

- using the newest EPRD Report: 

Grimberg A, Lützner J, Melsheimer O, Morlock M, Steinbrück A. EPRD Jahresbericht 2023 [Internet]. DE: EPRD Endoprothesenregister Deutschland; 2023 [cited 2024].

- Grammatical error in lines 320 – 321: 

“These results do not only apply to femurs of Dorr type A but to all Dorr categories, as in this study all Dorr types were used.”

- Figure numbering from figure 4 upwards was adapted to include the new figure 4. In old figure 7 and 8 the correlation coefficients were corrected to the ones in the text. All figures were reuploaded again with a higher quality without changes of the contents other than already mentioned.

---

## [Editor Report · Decision Letter 2]

7 Mar 2024

Small design modifications can improve the primary stability of a fully coated tapered wedge hip stem

PONE-D-23-33248R2

Dear Dr. Glismann,

We’re pleased to inform you that your manuscript has been judged scientifically suitable for publication and will be formally accepted for publication once it meets all outstanding technical requirements.

Kind regards,

Zhao Li, Ph.D., M.D.,

Academic Editor

PLOS ONE

---

## [Editor Report · Acceptance letter]

2 Apr 2024

PONE-D-23-33248R2 

PLOS ONE

Dear Dr. Glismann, 

I'm pleased to inform you that your manuscript has been deemed suitable for publication in PLOS ONE. Congratulations! Your manuscript is now being handed over to our production team.

Kind regards, 

on behalf of

Dr. Zhao Li 

Academic Editor

PLOS ONE